# Deciphering the catalytic mechanism of superoxide dismutase activity of carbon dot nanozyme

Wenhui Gao[1,9], Jiuyang He[2,9], Lei Chen[2,3,9], Xiangqin Meng[2], Yana Ma [1], Liangliang Cheng [1], Kangsheng Tu [4], Xingfa Gao [5], Cui Liu [1] ✉, Mingzhen Zhang [1,4] ✉, Kelong Fan [2,6,7] ✉, Dai-Wen Pang [8] ✉ & Xiyun Yan [2,6,7] ✉

Nanozymes with superoxide dismutase (SOD)-like activity have attracted increasing interest due to their ability to scavenge superoxide anion, the origin of most reactive oxygen species in vivo. However, SOD nanozymes reported thus far have yet to approach the activity of natural enzymes. Here, we report a carbon dot (C-dot) SOD nanozyme with a catalytic activity of over 10,000 U/mg, comparable to that of natural enzymes. Through selected chemical modifications and theoretical calculations, we show that the SOD-like activity of C-dots relies on the hydroxyl and carboxyl groups for binding superoxide anions and the carbonyl groups conjugated with the π-system for electron transfer. Moreover, C-dot SOD nanozymes exhibit intrinsic targeting ability to oxidation-damaged cells and effectively protect neuron cells in the ischemic stroke male mice model. Together, our study sheds light on the structure-activity relationship of C-dot SOD nanozymes, and demonstrates their potential for treating of oxidation stress related diseases.

The high catalytic efficiency and strong substrate specificity of natural enzymes make them ideal catalysts in biomedical applications. Horseradish peroxidase, for example, is frequently utilized in enzyme-based sensing for biomarkers[1], viruses[2], and bacteria[3]. Catalase, which catalyzes the decomposition of hydrogen peroxide into water and oxygen, could improve the antitumor efficiencies of radiotherapy[4], sonodynamic therapy[5], and photodynamic therapy[6] by relieving the hypoxia of tumor microenvironment. Superoxide dismutase has been used to treat skin inflammations[7], inflammatory arthritis[8], lung diseases and pulmonary fibrosis[9], and diabetic nephropathy[10], etc. However, most natural enzymes suffer from the ease of denaturation, high cost, difficulty of preparation, and being burdensome for mass production. To address these issues, artificial enzymes have emerged as stable and low-cost alternatives to enzymes[11]. Among which, nanomaterials with enzyme-like properties (nanozymes) have changed our perception of nanomaterials and enzyme mimics, gaining considerable attention due to their capacity to overcome the disadvantages of natural enzymes[12]. Over the past decade, many nanomaterials were

[1]School of Basic Medical Sciences, Xi'an Jiaotong University Health Science Center, Xi'an, Shaanxi 710061, P. R. China. [2]CAS Engineering Laboratory for Nanozyme, Key Laboratory of Protein and Peptide Pharmaceutical, Institute of Biophysics, Chinese Academy of Sciences, 100101 Beijing, P. R. China. [3]Laboratory of Theoretical and Computational Chemistry, Institute of Theoretical Chemistry, Jilin University, Changchun 130023, P. R. China. [4]Department of Hepatobiliary Surgery, the First Affiliated Hospital of Xi'an Jiaotong University, Xi'an, Shaanxi 710061, P. R. China. [5]National Center for Nanoscience and Technology, 100190 Beijing, P. R. China. [6]Nanozyme Medical Center, School of Basic Medical Sciences,  Zhengzhou University, Zhengzhou 450052, P. R. China. [7]University of Chinese Academy of Sciences, 101408 Beijing, P. R. China. [8]Haihe Laboratory of Sustainable Chemical Transformations, Tianjin Key Laboratory of Biosensing and Molecular Recognition, Frontiers Science Center for New Organic Matter, Research Center for Analytical Sciences, College of Chemistry, and Frontiers Science Center for Cell Responses, Nankai University, Tianjin 300071, P. R. China. [9]These authors contributed equally: Wenhui Gao, Jiuyang He, Lei Chen. ✉e-mail: liucui@xjtu.edu.cn; mzhang21@xjtu.edu.cn; fankelong@ibp.ac.cn; dwpang@whu.edu.cn; yanxy@ibp.ac.cn

designed to attain high enzyme-like activities by imitating natural enzyme active centers or integrating multivalent elements within nanostructures. However, because of the varying composition and structure of different nanozymes, many complicated catalytic behaviors have been observed in nanozymes, making it difficult to identify the active sites and, as a result, hard to design nanozymes with desirable catalytic activity and selectivity[13,14]. Up to now, nanozyme research mostly revolves around oxidoreductase activities, including oxidase (OXD)-[15], peroxidase (POD)-[16], catalase (CAT)-[17], and superoxide dismutase (SOD)-[18]like activities. Among them, POD nanozymes have been most extensively studied, from structure-activity relationship to catalytic mechanism, as well as their potential applications in disease diagnostics and therapy[19], with the activity of POD nanozyme, such as single atom nanozymes[20], now comparable to that of natural peroxidase enzymes. Aside from POD nanozymes, which catalyze the production of reactive oxygen species (ROS) in vivo[21], nanozymes with SOD-[22], CAT-[23], and glutathione peroxidase (GPx)-[24] like activities have been employed to eliminate ROS for cytoprotection, anti-inflammation, or antitumor theranostics[25–27]. However, research on SOD nanozymes, which catalyze the dismutation of superoxide radicals, the source of a vast majority of ROS in cells, is still minimal. Most reported SOD nanozymes typically exhibit modest catalytic efficiency (Supplementary Table 1), and the catalytic mechanism remains unclear. With the increasing number of disorders linked to ROS, clarifying the catalytic mechanism, and designing high activity SOD nanozymes are in urgent demand[28].

Carbon nanomaterials with well-defined electronic and geometric structures have shown the capability of mimicking the catalytic activities of natural enzymes. Carbon nanomaterials-based nanozymes are prospective alternatives for natural enzymes in biomedical applications due to their unique electrical, optical, thermal, and mechanical capabilities[29]. In particular, carbon dots (C-dots), as a class of photoluminescent nanomaterials, have received substantial interest in the past decade owing to their unique properties[30,31]. C-dots offer the benefits of small size, easy synthesis, cheap cost, controllable luminescence emission, strong photostability, and superior biocompatibility over other fluorescent nanomaterials such as quantum dots, metal clusters, and rare earth nanoparticles[32–34]. Moreover, abundant oxygen-containing functional groups on the surface of C-dots, such as carbonyl, carboxyl, hydroxyl, and other functional groups, endow them with good water solubility and ease of functionalization[35–37]. Consequently, C-dots have shown great potential applications in sensing, bioimaging, light-emitting diodes, therapy, etc[38–40]. Of note, C-dots exhibit catalytic activity due to their size effect and abundant active sites[41,42]. C-dots with peroxidase-like activity were synthesized via the oxidation of candle soot by Zheng et al.[43]. Since then, the C-dots with enzyme-like activity have gained considerable scientific interest and have been effectively employed in the detection of hydrogen peroxide[44], cholesterol[45], glucose[46,47], carcinoembryonic antigen[48], as well as in tumor imaging[49], cell detoxification[50], anti-inflammation[51], and cancer therapy[52,53]. Previous works mainly focused on the peroxidase activity of C-dots, while reports on how to design C-dot nanozymes with high antioxidant activity are scarce.

In this work, taking advantage of the tunable surface functional groups and well-defined structure of carbon material, we design C-dots with high SOD-like activity (C-dot SOD nanozyme, >10,000 U/mg) and unveil their catalytic mechanism. Surface structure tuning and theoretical calculations reveal the surface state-related catalytic activity of C-dot SOD nanozyme. The hydroxyl and carboxyl groups of the C-dots bind superoxide anions and the carbonyl groups oxidize superoxide anions, producing oxygen and reduced-state C-dots. The reduced C-dots are oxidized back into the initial state by another superoxide anion and produce hydrogen peroxide ($H_2O_2$). Importantly, the C-dot nanozymes selectively target oxidation-damaged cell and mitochondria. Combined with its high catalytic activity, we successfully employ C-dot SOD nanozymes to reduce the intracellular ROS level and protect neurons from oxidation stress caused by ischemic stroke in vivo. Moreover, C-dot SOD nanozymes possess the advantages of high stability, facile preparation, low cost, and easy-to-scale production, overcoming the limitations of natural enzymes and showing great application potential in industrial, medical, and biological fields, etc.

## Results and discussions

### Preparation and characterization of C-dot SOD nanozymes

We first set out to determine the ideal method to synthesize C-dots with high SOD-like activity. C-dots were synthesized from larger carbon structures, i.e. graphite powder, carbon black, and activated charcoal, by oxidative treatment with a mixture of nitric acid and sulfuric acid ($V_{HNO_3} : V_{H_2SO_4} = 1 : 1$). Transmission electron microscopy (TEM) images (Fig. 1a–c) showed that the C-dots derived from graphite powder, carbon black, and activated charcoal were homogeneous with average diameter of $2.3 \pm 0.4$, $2.1 \pm 0.4$, and $2.0 \pm 0.4$ nm, respectively. As shown in the high-resolution TEM image (Fig. 1a–c inset), all of the C-dots exhibited crystallinity with lattice spacings of 0.32, 0.34 and 0.21 nm, corresponding to the (002), (002) and (100) facets of graphite, respectively[54,55]. The SOD-like activity of the C-dots was then quantified using a commercial SOD assay kit (WST-1). As shown in Fig. 1d, the SOD-like activity was represented by the enzyme specific activity (U/mg). C-dots synthesized from graphite powder and carbon black showed relatively low activities of 405 and 418 U/mg, respectively. Surprisingly, the SOD-like activity of C-dots prepared from activated charcoal exhibited an ultrahigh SOD-like activity of $1.1 \times 10^4$ U/mg, which was substantially greater than the SOD-like activity of already reported SOD nanozymes and even natural SOD enzyme (Supplementary Table 1). This could be due to the differences in structural features of these raw carbon materials, such as crystal structure (Supplementary Fig. 1), density, porosity, hardness, etc., which have effect on the thermodynamics and kinetics of the oxidation and etching process that affect the formation, surface structure, and catalytic performance of C-dots.

To investigate the determining factor for the SOD-like activity of C-dots, the surface structural differences of the C-dots derived from the three kinds of materials were investigated. The Raman spectra (Fig. 1e) of these C-dots displayed the G- (1596 cm$^{-1}$) and D- (1380 cm$^{-1}$) bands with $I_D/I_G$ of 0.9 to 1.0, indicating large portion of defects on their surface induced by the strong oxidation. The XRD patterns of these C-dots are shown in Fig. 1f. The diffraction peaks of activated charcoal- and carbon black-derived C-dots with 2θ values of 25-26 ° and 42-46 ° attributing to the (002) and (100) facets, respectively, of graphite [powder diffraction file (PDF Card No. 01-0640)]. The (002) and (100) facets correspond to the facets parallel and perpendicular to the sp²-carbon layer of graphite, respectively, consisting with the TEM results. In contrast, no discernible diffraction peaks were detected in graphite powder-derived C-dots, which may be due to the severe structural damages during the oxidation process. C 1s X-ray photoelectron spectroscopy (XPS) was conducted to semi-quantitatively analyze the surface structures of these C-dots (Fig. 1g–i). The XPS results indicated the presence of C = C, C − O, C = O, and O − C = O on the surface of these C-dots. The carbon-to-oxygen ratios of graphite powder-, and carbon black-derived C-dots were 1.33 and 1.45, respectively, much lower than that of activated charcoal-derived C-dots (2.01), indicating that graphite powder-, and carbon black-derived C-dots possess higher degree of surface oxidation. The C = C content of activated charcoal-derived C-dots was as high as 71%, while those of graphite powder- and carbon black-derived C-dots were only 57% and 64%, respectively (Supplementary Table 2). The high content of C = C suggests a large π-electron system that could promote electron transfer and stabilize intermediate products containing unpaired electrons. Therefore, sufficient C = C content is necessary for C-dots with high SOD enzymatic activity.

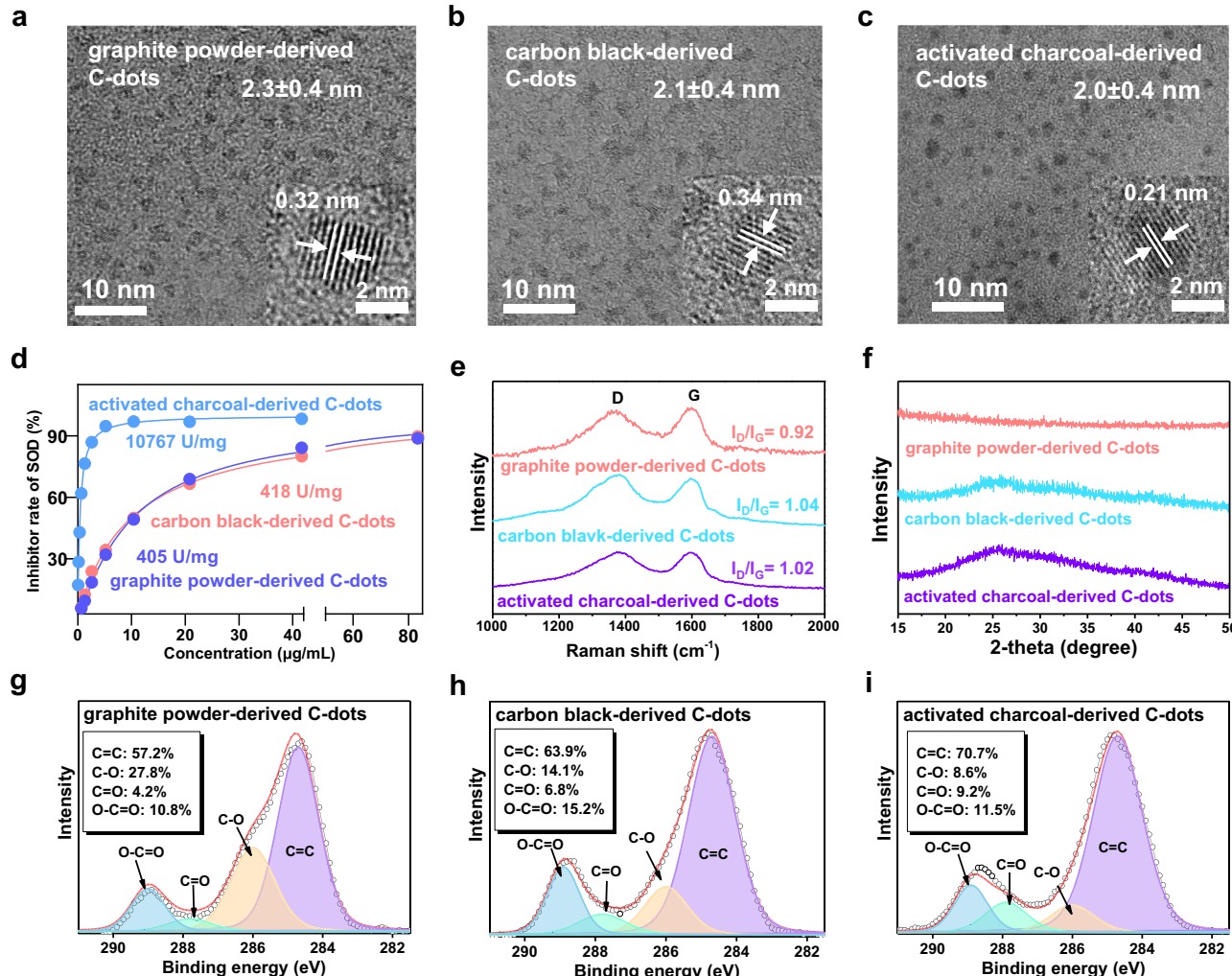

**Fig. 1 | Characterization of C-dot SOD nanozymes.** TEM (Inset: HR-TEM) images of C-dots prepared from (**a**) graphite powder, (**b**) carbon black, and (**c**) activated charcoal. **d** The SOD-like activities of C-dots prepared from graphite powder, carbon black, and activated charcoal. **e** Raman spectra and (**f**) XRD patterns of C-dots prepared from graphite powder, carbon black, and activated charcoal as indicated. C 1s high-resolution XPS spectra with identification of peaks by curve fitting of C-dots prepared from (**g**) graphite powder, (**h**) carbon black, and (**i**) activated charcoal. For (**a**–**c**), three times each experiment were repeated independently with similar results. In (**d**), data are presented as means ± SD from three independent experiments.

In addition, we also found that compared to activated charcoal-derived C-dots, carbon black-derived C-dots and graphite powder-derived C-dots exhibited lower carbonyl content, suggesting that carbonyl groups may also affect the SOD-like activity of C-dots. In the FT-IR spectra of these C-dots (Supplementary Figs. 2 and 3), the strong bands at 3412, 1726, and 1240 cm$^{-1}$ were ascribed to the stretching vibration of O-H, C = O, and C-O, respectively. The absorption bands at 1620 and 1350 cm$^{-1}$ could be attributed to the stretching vibration of C = C and the bending vibration of C-H, respectively. The peaks ranging from 2870 to 2980 cm$^{-1}$ were attributed to the stretching vibration of C-H in aliphatic hydrocarbons while the broadband around 2560 cm$^{-1}$ was attributed to hydrogen bond stretching vibration[38,56,57]. The significant difference in absorption peak intensities of these functional groups among the three kinds of C-dots indicated a variation in the contents of functional groups. Surface functional groups of C-dots could be quantified by $^1$H-NMR spectroscopy using potassium biphthalate (PBP) as an internal standard[56] (Fig. 2d and Supplementary Fig. 4). The total content of the reactive carboxyl and hydroxyl groups on graphite powder-, and carbon black-derived C-dots were calculated to be 0.85 and 3.08 mmol/g, respectively, lower than that of activated charcoal-derived C-dots (4.35 mmol/g). However, XPS results showed that the contents of C-O and O-C = O of graphite powder-, and carbon

black-derived C-dots were higher than that of activated charcoal-derived C-dots, not consistent with the quantitative analysis of $^1$H-NMR. The reason was that C-O and O-C = O on the surface of graphite powder and carbon black-derived C-dots were less reactive. This could be due to their strong steric effects or other existence forms, including ethers and esters, which XPS was unable to discriminate from hydroxyl and carboxyl, respectively.

From the above results, it is reasoned that surface oxygen-containing groups play a key role in the catalytic activity of C-dots. The oxidation etching in the synthesis process destructs the relatively complete π-electron system of the original carbon materials, inducing oxygen-containing functional groups, such as carboxyl, hydroxyl, and carbonyl groups, on the surface of the C-dots. Simultaneously, the initially ordered sp$^2$ network structure of raw materials converted into the sp$^2$-sp$^3$ hybrid nanostructure. Finally, the small size (~2 nm) and large specific surface area enable C-dot SOD nanozyme to provide abundant binding and catalytic sites for the catalytic reaction. Oxygen-containing functional groups would combine with superoxide anions through weak interactions such as static electricity, hydrogen bonds, and other van der Waals forces, etc., facilitating redox reactions. Moreover, activated charcoal-derived C-dots synthesized with 0.5 h and 2 h reaction times exhibited lower SOD-like activities than the

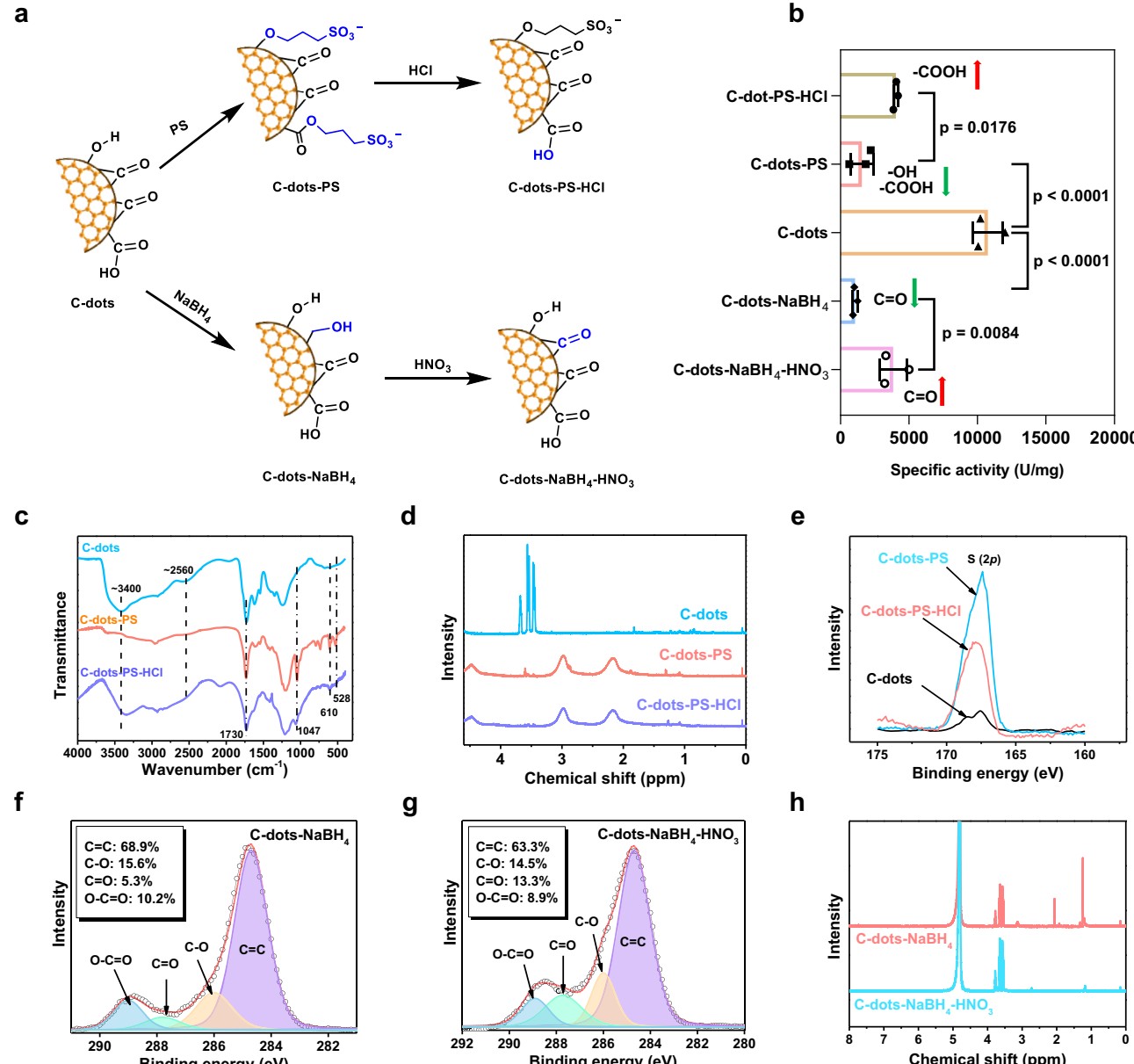

**Fig. 2 | Surface modifications to determine the catalytic active site of C-dot SOD nanozymes. a** Illustration of C-dots modification. **b** SOD-like activity change of C-dots before and after passivation, reduction, and re-oxidation. **c** FT-IR, (**d**) ${}^1$H NMR, and (**e**) S 2$p$ XPS spectra of C-dots, C-dots-PS and C-dots-PS-HCl. C 1$s$ high-resolution XPS spectra with identification of peaks by curve fitting of (**f**) C-dots-NaBH$_4$ and (**g**) C-dots-NaBH$_4$-HNO$_3$. **h** ${}^1$H NMR spectra of C-dots-NaBH$_4$ and C-dots-NaBH$_4$-HNO$_3$. In (**b**), $P$ values are determined with one-way ANOVA Tukey's multiple comparisons test. Data are presented as means ± SD from three independent experiments. Source data are provided as a Source Data file.

optimal 1.5 h (Supplementary Fig. 5), suggesting the reaction time in synthesis affected the SOD-like activity of the prepared C-dot nanozyme. Our previous work demonstrated that the surface-oxidation degree of C-dots increases as the reaction time prolong[24], which confirmed the surface-related SOD-like activity of C-dot nanozymes.

## Deciphering the mechanism of SOD-like activity of C-dot nanozymes

Due to the excessive number of variables in a series of samples produced under various reaction circumstances or utilizing different raw materials, the inferred catalytic mechanism is unlikely to be convincing enough. To better understand the origin of C-dot nanozyme activity, we utilized the C-dots with the greatest enzymatic activity, which were produced from activated charcoal with an ideal reaction time of 1.5 h, as a model for investigation.

First, we tested whether C-dots also possessed other enzyme-like activities in addition to SOD-like activity, such as catalase-, peroxidase- and oxidase-like activities. Monitoring the decomposition of H$_2$O$_2$ was used to quantify catalase-like activity. By measuring the oxidation of 3,3′,5,5′-tetramethyl-benzidine (TMB) in the presence of H$_2$O$_2$ and dissolved oxygen, respectively, peroxidase- and oxidase-like activities were detected. As shown in Supplementary Fig. 6, no significant catalase, peroxidase, or oxidase-like activities were detected in the C-dots. As a result of the relatively exclusive SOD-like activity of as-prepared C-dots, a more detailed examination into the mechanism of their catalytic activity is possible.

To investigate the role of the carboxyl and hydroxyl groups on the surface of C-dots in SOD-like activity, we selectively used 1,3-propanesultone (PS) to deactivate carboxyl and hydroxyl groups. 1,3-propanesultone (PS) reacts with carboxyl and hydroxyl groups on

the surface of C-dots to form ester and ether, respectively (indicated as C-dots-PS), as previously reported in our work[56,57]. Under acidic circumstances, the ester can hydrolyze, while ethers cannot. Thus, following the hydrolysis of C-dots-PS in 0.1 M HCl solution, C-dots with only the hydroxyl groups passivated, C-dots-PS-HCl, were obtained (Fig. 2a). FT-IR and [1]H NMR spectra of C-dots before and after the reactions were recorded to confirm the surface modification. As shown in Fig. 2c, the O-H absorption band around 3400 cm[−1] decreased in the FT-IR spectrum of C-dots-PS and increased in that of C-dots-PS-HCl. After carboxyl groups were converted into esters, the absorption intensity of C = O increased. The peak around 1730 cm[−1] increased in the FT-IR spectrum of C-dots-PS while the peak in C-dots-PS-HCl decreased. In addition, the characteristic absorption peaks of the sulfonic acid group ($-SO_3^-$) at 528, 610, and 1047 cm[−1] appeared in the FT-IR spectrum of C-dots-PS and decreased in that of C-dots-PS-HCl, while the stretching vibration of the hydrogen bond of associating carboxyl group (2560 cm[−1]) decreased in the FT-IR spectrum of C-dots-PS and restored in that of C-dots-PS-HCl. In the [1]H NMR spectrum of C-dots-PS (Fig. 2d), three new peaks at 2.1, 2.9, and 4.4 ppm appeared, which were corresponding to [1]H nuclei of β, α, and γ of $-SO_3^-$, respectively[57]. These characteristic absorption peaks decreased significantly in the [1]H NMR spectrum of the C-dots-PS-HCl. Moreover, the XPS spectra showed that the sulfur content increased in C-dots-PS and decreased in that of C-dots-PS-HCl (Fig. 2e). These results indicated the successful modification of the C-dots surface.

The SOD-like activity of C-dots-PS was determined to be $1.6 \times 10^3$ U/mg (Fig. 2b), indicating that the passivation of hydroxyl and carboxyl groups decreases the SOD-like activity of C-dots. For C-dots-PS-HCl, in which the carboxyl groups recovered while hydroxyl groups remained passivated, the SOD-like activity upturned to $4.0 \times 10^3$ U/mg. To rule out the possibility of the hydrolysis condition increasing the nanozyme activity of C-dots, the original C-dots were refluxed in 0.1 M HCl for 12 h (the same procedure as hydrolysis), and the resulting samples was denoted as C-dots-HCl. The SOD-like activity of C-dots-HCl decreased slightly (Supplementary Fig. 7), indicating that the hydrolysis process does not increase the activity of C-dots. As a result, the increase in catalytic activity of C-dots-PS-HCl compared to that of C-dots-PS could be reliably attributed to the carboxyl recovery. Moreover, the SOD-like activity of C-dots-PS-HCl was much lower than that of C-dots-HCl, indicating that hydroxyl also plays a key role in the SOD-like activity of C-dots. Therefore, both carboxyl and hydroxyl groups are important structure in the catalytic activity site of C-dots.

To study the contribution of the carbonyl groups to the nanozyme activity of C-dots, we utilized sodium borohydride ($NaBH_4$) to reduce the carbonyl groups on the surface of C-dot SOD nanozyme (Fig. 2a), as previously reported in our work[58]. The XPS spectrum of reduced C-dots (C-dots-$NaBH_4$) showed that the peak attributing to C-O increased significantly while the peak attributing to C = O decreased (Fig. 2f). The content ratio of hydroxyl to carbonyl increased from 0.9 (C-dots) to 2.9 (C-dots-$NaBH_4$). In the [1]H NMR spectrum of C-dots-$NaBH_4$, the peaks ranging from 3.38 to 3.74 ppm decreased, suggesting that the carbonyl groups are reduced to hydroxyl groups. The peaks ranging from 1.05 to 2.00 ppm increased (Fig. 2h), which can be attributed to β-H of the newly formed hydroxyl group converted from the carbonyl group[58]. The SOD-like activity of C-dots-$NaBH_4$ was determined to be $1.1 \times 10^3$ U/mg (Fig. 2b), which was much lower than that of C-dots, indicating that the carbonyl group plays a critical role in the catalytic activity site. To verify this, we used 5 M $HNO_3$ to oxidize the C-dots-$NaBH_4$ for 36 h at 40 °C, producing C-dots-$NaBH_4$-$HNO_3$. The XPS spectrum of C-dots-$NaBH_4$-$HNO_3$ showed that the peak of C-O decreased while the peak of C = O increased (Fig. 2g). The ratio of hydroxyl to carbonyl decreased to 1.1, indicating that the hydroxyl groups are oxidized to C = O. In [1]H NMR of C-dots-$NaBH_4$-$HNO_3$, the peaks ranging from 1.05 to 2.00 ppm decreased to a negligible level, also confirming the re-oxidation of C-dots-$NaBH_4$ (Fig. 2h). The SOD-

like activity of producing C-dots-$NaBH_4$-$HNO_3$ was restored to $3.8 \times 10^3$ U/mg (Fig. 2b). The reason for the incomplete recovery of SOD-like activity of C-dots-$NaBH_4$-$HNO_3$ may be due to the decrease of C = C content caused by the oxidation process.

To further study the surface state-related nanozyme activity, the C-dots were hydrothermally treated in a NaOH solution (5 M) for 24 h at 200 °C, and subsequently were reduced by hydroiodic acid (HI) in acetic acid[38,58], producing C-dots-NaOH-200 °C and C-dots-NaOH-HI (Fig. 3a). For C-dots-NaOH-200 °C, carbonyl content decreased from 9.2% to 3.7%, C = C content increased from 71% to 74%, while the contents of hydroxyl and carboxyl did not change significantly (Fig. 3b). To confirm the effect of π-system on the SOD-like activity, the C-dots were stirred in a 0.5 M NaOH solution for 24 h at 40 °C (Fig. 3a), producing C-dots-NaOH-40 °C. As shown in Fig. 3c, the carbonyl group content of C-dots-NaOH-40 °C decreased to 4.4%, while the C = C content almost stayed unchanged. The $I_D/I_G$ for C-dots-NaOH-40 °C was 1.01 (Fig. 3d), which was similar to that of C-dots (1.02), confirming that the conjugated system remained unchanged. For C-dots-NaOH-HI, almost all the carbonyl groups were removed, while C = C content increased to 79.8% (Fig. 3e). The $I_D/I_G$ for C-dots-NaOH-HI (0.71) was much smaller than that of C-dots-NaOH-200 °C (0.97) (Fig. 3d). The SOD-like activity of C-dots-NaOH-HI decreased to 100 U/mg, 1% of C-dots (Fig. 3f), indicating that the carbonyl group may be the most important factor for the SOD-like activity of C-dots. The SOD-like activity of C-dots-NaOH-200 °C decreased to 810 U/mg (8% of C-dots) while that of C-dots-NaOH-40 °C decreased to 448 U/mg (4% of C-dots) (Fig. 3f). Considering the contents of hydroxyl, carbonyl and carboxyl groups of C-dots-NaOH-200 °C were lower than that of C-dots-NaOH-40 °C, the lower SOD-like activity of C-dots-NaOH-40 °C could be attributed to the lower C = C content, confirming a key role of π-system of C-dots for their SOD-like activity (Supplementary Table 2).

Our above experimental evidences indicate that, although there are many factors affecting the SOD-like activity of C-dot nanozyme, the carbonyl groups of C-dot SOD nanozyme are the catalytic site of the SOD-like activity and the hydroxyl groups play an important role in binding with the reactants. Based on the experimental results, we simulated the SOD-like activity by a carbonyl group of C-dot nanozyme with and without hydroxyl groups to verify the catalytic mechanism. The proposed reaction pathway of the SOD-like catalytic cycle was investigated by employing density functional theory (DFT) calculations as shown in Fig. 4. As is known that the $O_2^{·-}$ is reported to be a Brønsted base with pKb = 9.12[59], and it can readily capture a proton from water solution forming $HO_2^·$ and $OH^-$, as illustrated by the following equation:

$$O_2^{·-} + H_2O \rightarrow HO_2^· + OH^- \tag{1}$$

Therefore, the following below equations serve as a plausible mechanism for the SOD-like activity of a catalytic site:

$$2HO_2^· \rightarrow O_2 + H_2O_2 \tag{2}$$

$$HO_2^· + O = (C - dot) = O \rightarrow O_2 + HO - (C - dot) - O^· \tag{3}$$

$$HO_2^· + HO - (C - dot) - O^· \rightarrow H_2O_2 + O = (C - dot) = O \tag{4}$$

The whole SOD-like catalytic cycle (Eq. (2)) of a catalytic site consists of two elementary reactions including an oxidation reaction (Eq. (3)) and a reduction reaction (Eq. (4)). A carbonyl group of C-dot (named π-C = O) can be converted to a hydroxyl group (named π-OH) by oxidizing a $HO_2^·$ free radical to produce an $O_2$ molecule (from state 2 to state 3 in Fig. 4a and c), and a hydroxyl group of C-dot can be converted to a carbonyl group by reducing a $HO_2^·$ free radical to

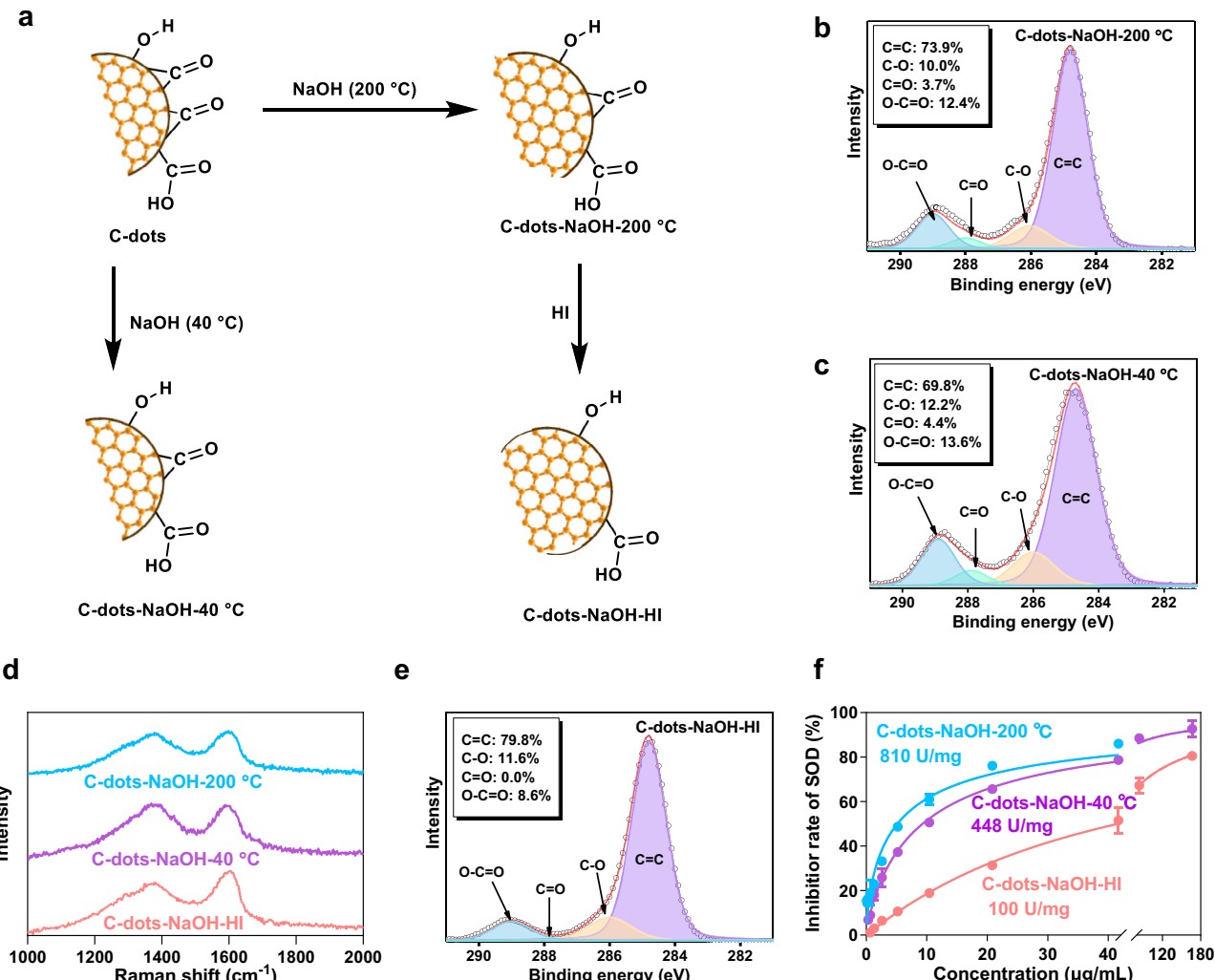

**Fig. 3 | Surface modifications to determine the catalytic active site of C-dot SOD nanozymes. a** Illustration of C-dots modification. C 1 *s* high-resolution XPS spectra with identification of peaks by curve fitting of (**b**) C-dots-NaOH-200 °C, (**c**) C-dots-NaOH-40 °C and (**e**) C-dots-NaOH-HI. **d** Raman spectra of C-dots-NaOH-200 °C, C-dots-NaOH-40 °C and C-dots-NaOH-HI. **f** SOD-like activities of C-dots-NaOH-200 °C, C-dots-NaOH-40 °C, and C-dots-NaOH-HI. In (**f**), data are presented as means ± SD from three independent experiments.

produce an $H_2O_2$ molecule (from state 4 to state 4 in Fig. 4a and c). Figure 4a and c indicates that there are both seven different states including 2 transition states (named TS1/TS2) in the proposed reaction pathway for the bare $\pi$-C = O group and hydroxyl assisted $\pi$-C = O group to achieve the SOD-like catalytic cycle. Our calculations show that, compared with the C-dot nanozyme without hydroxyl group, the binding energy between $HO_2^{\cdot}$ and C-dot nanozyme with hydroxyl group is much lower (−0.65 eV vs. −0.54 eV as shown in Fig. 4c and d), which indicates that the hydroxyl C-dots have stronger ability to capture $HO_2^{\cdot}$ free radicals. In the Michaelis-Menten equation, $K_M$ denotes the equilibrium constant of the following reaction:

$$HO_2^{\cdot *} \rightarrow HO_2^{\cdot} + {}^*, \Delta G_m^o \qquad (5)$$

Where the asterisk (*) denotes an unoccupied active site on the C-dot and the $\Delta G^o_m$ denotes the change of standard Gibbs free energy. According to the van't Hoff equation,

$$K_M^o = e^{\Delta G_m^o / RT} \qquad (6)$$

Where $K^o_M$ denotes the standard dissociation constant, R denotes the gas constant, and T denotes the temperature. According to the above formula, the ratio for two $K^o_M$ values of SOD-like catalytic cycles of

C-dot nanozymes with and without hydroxyl group was about 1.4/100, which indicates the SOD-like catalysis was much easier to occur for hydroxyl assisted $\pi$-C = O groups than that of bare $\pi$-C = O groups. This calculation result was generally consistent with the experimental finding that the passivation of hydroxyl groups reduces the SOD-like activity of C-dots to ~16%. These results indicate that hydroxyl is the key structure of the SOD-like catalytic activity site of the C-dot nanozyme. This mechanism study shows that the SOD-like activity of C-dot nanozyme relies on the functional groups, and this understanding enables the rational design of C-dot SOD nanozyme with higher activity.

Moreover, C-dot SOD nanozymes showed robust thermal, acidic, and alkali stability (Supplementary Fig. 8a). After pretreatment in 37, 60, and 90 °C water bath for 1 h, the C-dot SOD nanozymes still retained high SOD-like activities at 90–95% of the original activity, indicating that C-dot SOD nanozyme possesses better thermal stability than the natural SOD, which denatures in high temperature[60]. The C-dot SOD nanozymes were also pretreated with hydrochloric acid (HCl) and sodium hydroxide (NaOH) solution to study their resistance to acid and alkali. After pretreatment in 0.05 M HCl and 0.01 M NaOH solutions, the C-dot SOD nanozymes still retained SOD-like activities of 92% and 84% (Supplementary Fig. 8b), respectively, which are unrealizable in natural SOD. Electron spin resonance (ESR) spectroscopy was

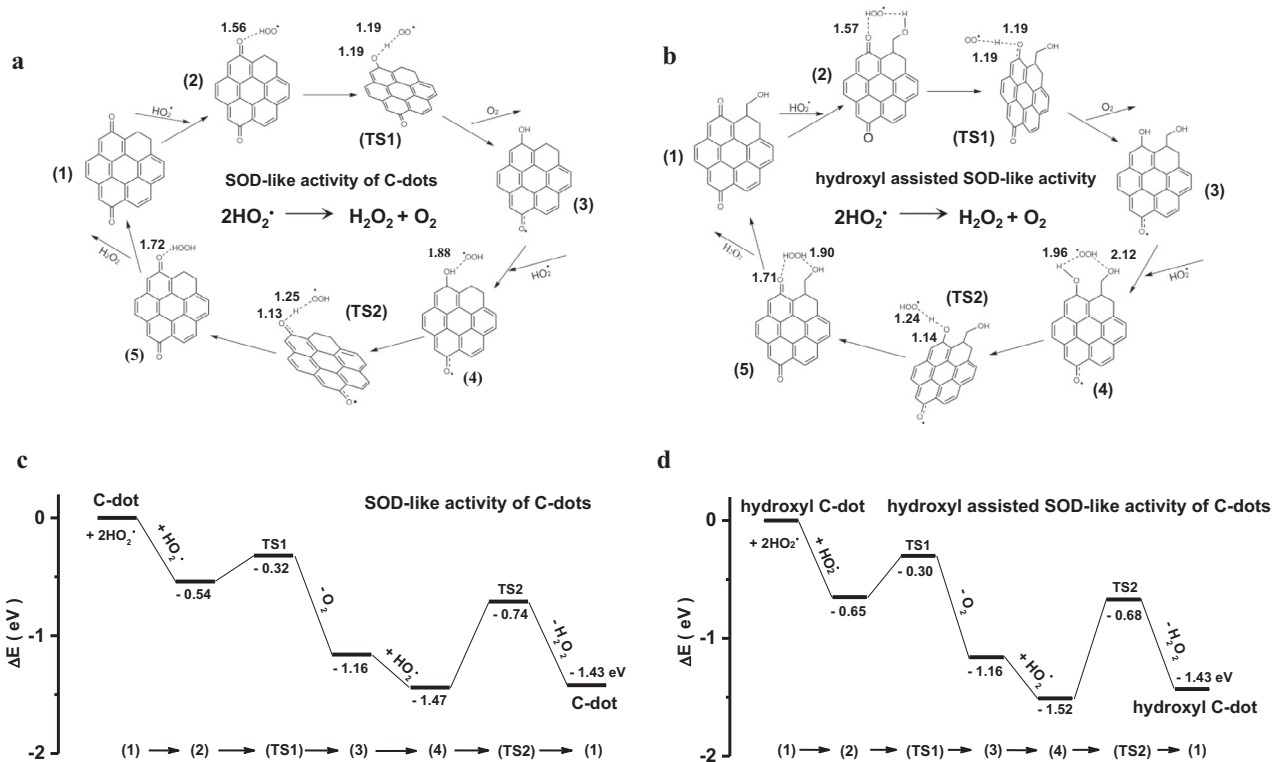

**Fig. 4 | Proposed SOD-like activity of C-dot nanozyme with and without hydroxyl groups. a** Proposed reaction pathway to achieve the SOD-like catalytic cycle of C-dot nanozyme without hydroxyl groups. **b** Proposed reaction pathway to achieve the SOD-like catalytic cycle of C-dot nanozyme with hydroxyl groups. **c** Gibbs free energy profile for a SOD-like catalytic cycle of C-dot nanozyme without hydroxyl groups. **d** Gibbs free energy profile for a SOD-like catalytic cycle of C-dot nanozyme with hydroxyl groups.

used to assess the superoxide radical ($O_2^{\cdot-}$)-scavenging capacity of C-dot SOD nanozyme. Superoxide generated from the reaction of L-methionine with riboflavin under LED irradiation could be trapped by 5,5-dimethyl-1-pyrroline-N-oxide (DMPO), producing the adduct of DMPO/$^\cdot$OOH. The ESR signal of DMPO/$^\cdot$OOH decreased along with the increase of C-dot SOD nanozyme concentration, which indicated that C-dots exhibit SOD-like activity and directly eliminate $O_2^{\cdot-}$ (Supplementary Fig. 9).

## C-dot SOD nanozymes specifically targeting the ROS-damaged cells

Given their remarkable SOD-like activity, C-dot SOD nanozymes have potential to protect cells against excessive ROS. During reperfusion after ischemic stroke, the level of ROS rapidly increases to a high point that typically induce the breakdown of the blood-brain barrier, the death of neuron cells and cascade inflammations. Here, the intracellular antioxidant capacity of C-dot SOD nanozymes was investigated with the intent of using the C-dot SOD nanozymes to scavenge ROS caused by reperfusion after ischemic stroke.

To assess whether C-dot SOD nanozymes are more prone to enter cells with high ROS level (Fig. 5a), we incubated Cyanine-5.5 (Cy5.5) labeled C-dots (C-dots-Cy5.5) with human neuroblastoma (SH-SY5Y) cells treated with $H_2O_2$ (50 μM, 100 μM, and 200 μM). As shown in Fig. 5b, C-dot SOD nanozymes tended to accumulate more in the cells treated with higher concentration of $H_2O_2$. The accumulation of C-dots in the cells was further quantified by using flow cytometry, and the results show that C-dots accumulated significantly higher in 200 M $H_2O_2$ treated cells than in PBS treated cells (Fig. 5c, e). These results indicate that the targeting ability of C-dot SOD nanozymes to oxidation-damaged cells, which may be due to the better permeability of membrane damaged by oxidative stress, is

consistent with the previous report that graphene-based nanoparticles target tumor cells by a cell membrane permeability targeting mechanism[61]. One of the challenges with SOD nanozymes used in vivo is that many nanoparticles are trafficked into endo/lysosomes after cell uptake, and thus lysosomes are a huge barrier against efficient SOD-like activity, since the pH levels in lysosomes are not ideal for the SOD nanozyme. Therefore, subcellular location of the nanozyme is also a key factor for determining its enzyme activity and ability to scavenge free radicals. Mitochondria are the primary organelles for the generation of $O_2^{\cdot-}$ that induce oxidative damage of cells. Following the staining of mitochondria with Mitotracker, we found that C-dot SOD nanozymes showed mitochondrial accumulation (Fig. 5d) with a Pearson's correlation coefficient of 0.43[62]. Furthermore, co-staining C-dots with lysosome showed no visible overlap (Fig. 5f) with a Pearson's correlation coefficient of 0.03, indicating no significant accumulation of C-dot SOD nanozyme in the lysosome. These results demonstrated that the C-dot SOD nanozymes are capable of targeting the mitochondria by overcoming the cell membrane.

Next, we evaluated whether C-dot SOD nanozymes can reduce intracellular ROS. SH-SY5Y and RAW264.7 (mouse peritoneal macrophages) cells were cultured to assess the cytotoxicity of C-dot SOD nanozymes, and to study the potential of C-dot SOD nanozymes in scavenging ROS in vitro. 2′,7′-dichlorodihydrofluorescein diacetate (DCFH-DA) was employed as a probe for the detection of intracellular ROS. SH-SY5Y were treated with paraquat, a widely used reagent to stimulate $O_2^{\cdot-}$ production in cells, to increase the intracellular ROS level. The fluorescence intensity of the cells, as shown in Fig. 5g, indicates the ROS level. Cells treated with C-dot SOD nanozymes together with paraquat exhibited lower fluorescence intensity than cells treated with paraquat alone. Using flow cytometry, we further quantified the

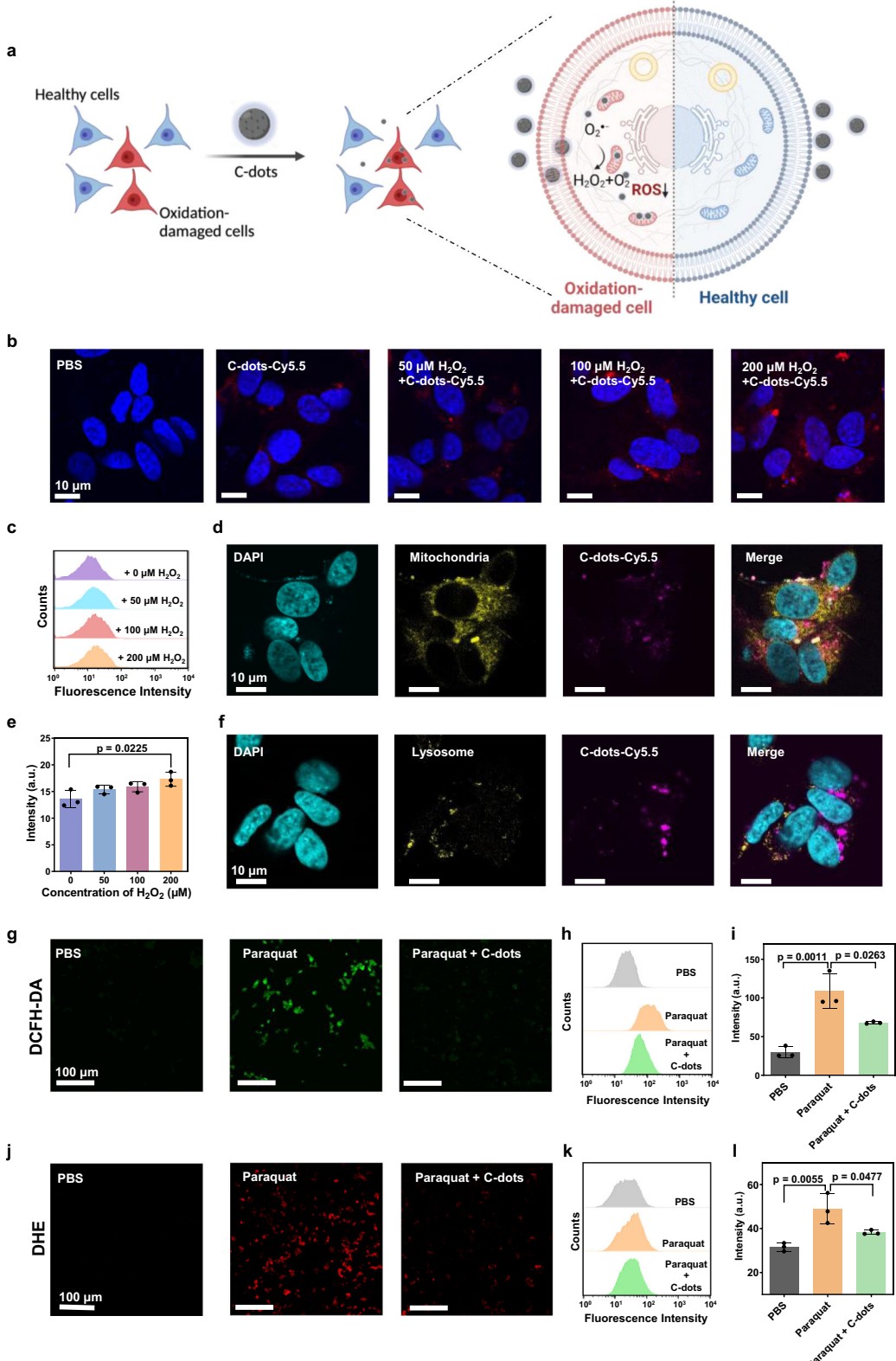

ROS level of the cells (Fig. 5h). The results show that cells co-incubated with paraquat and C-dot SOD nanozymes exhibited significantly lower ROS level than that of paraquat treated cells (Fig. 5i), indicating that the C-dot SOD nanozymes have the capacity to reduce ROS levels in living cells. Dihydroethidium (DHE) was used to assess the ability of C-dot to eliminate $O_2^{\cdot-}$ in living cells. As shown in Fig. 5j, cells treated by C-dot SOD nanozymes together with paraquat exhibited a reduced level of fluorescence, and the flow cytometry results confirmed that C-dot SOD nanozymes treated group exhibits significantly lower fluorescence (Fig. 5k and l). The cell viability also showed that co-incubation with C-dot SOD nanozymes significantly decreased the cell death caused by the paraquat (Supplementary Fig. 10). These results indicated that the C-dot SOD nanozymes successfully protect living cells from ROS through scavenging $O_2^{\cdot-}$, the origin of ROS.

**Fig. 5 | Oxidation-damaged cell targeting ability and ROS scavenging capacity of C-dot SOD nanozymes. a** Illustration of the selective targeting and ROS scavenging ability of C-dot SOD nanozymes to oxidation-damaged cells, adapted from "Compare and Contrast Layout−Cell", by BioRender.com (2023). Retrieved from https://app.biorender.com/biorender-templates. **b** Representative confocal imaging of C-dot SOD nanozymes accumulating in SH-SY5Y cells in the presence of various concentrations of $H_2O_2$ ($n = 3$ independent experiments). **c** Flow cytometry analysis and (**e**) corresponding quantification analysis of mean fluorescence density for the accumulation of C-dot SOD nanozyme in SH-SY5Y cells ($n = 3$ independent experiments). **d** Confocal images of the colocalization of Cy5.5 labeled C-dot SOD nanozymes (magenta) with mitochondria (yellow), and nuclei stained with DAPI (cyan) ($n = 3$ independent experiments). **f** Confocal images of the colocalization of Cy5.5 labeled C-dot SOD nanozymes (magenta) with lysosome (yellow), and nuclei stained with DAPI (blue) ($n = 3$ independent experiments). **g** ROS production detected by fluorescence probe DCFH-DA in SH-SY5Y cells using confocal microscopy. **h** Flow cytometry analysis and (**i**) corresponding quantification analysis of ROS levels in cells with different treatments ($n = 3$ independent experiments). **j** $O_2^{\cdot-}$ concentration detected by fluorescence probe DHE in SH-SY5Y cells using confocal microscopy. **k** Flow cytometry analysis and (**l**) corresponding quantification analysis of $O_2^{\cdot-}$ levels in cells with different treatments ($n = 3$ independent experiments). In (**e**, **i**, and **l**), data are presented as means ± SD from 3 independent experiments. *P* values are determined with one-way ANOVA Tukey's multiple comparisons test. Source data are provided as a Source Data file. For (**c**, **h**, and **k**), the gate strategies were shown in Source Data.

The ischemic microenvironment also activates macrophage or microglia that already lie resident in the infarcted area. To test whether C-dot SOD nanozymes were also capable of eliminating ROS from macrophages, RAW264.7 cells were incubated with C-dot SOD nanozymes and then treated with exogenously Rosup. The intracellular ROS and $O_2^{\cdot-}$ were determined using DCFH-DA and DHE as the probes, respectively. As shown in Supplementary Fig. 11, negligible fluorescence signal was detected in the control group (untreated cells) and the cells incubated with C-dot SOD nanozyme, confirming good biocompatibility of the C-dot SOD nanozyme. While high fluorescence intensity was detected in cells treated with Rosup, with the assistance of C-dot SOD nanozymes, the fluorescence intensity of the cells treated with Rosup significantly decreased, indicating the antioxidant ability of C-dot SOD nanozymes. Overall, these results demonstrated the ROS scavenging capacity of C-dot SOD nanozymes in living cells.

### C-dot SOD nanozymes alleviating neurological damage induced by ischemic stroke

Ischemic stroke is one of the main causes of death globally, with considerable morbidity, disability, recurrence, and mortality. However, current treatments for ischemic stroke are far from satisfactory[63]. Part of the reason is that ROS are elevated in the ischemic region after the reperfusion, leading to the oxidative damage to neurons[64]. Therefore, ROS scavenging nanozymes have the potential in relieving the pathological conditions[65].

Here, to determine whether C-dot SOD nanozyme can successfully scavenge ROS in vivo, a typical middle cerebral artery occlusion (MCAO) infarcted ischemic stroke mice model was employed. We first investigated the half-life and pharmacokinetic behavior of C-dot SOD nanozyme in vivo by monitoring the fluorescence of Cy5.5 labeled on C-dot nanozymes. As shown in Fig. 6a, C-dot nanozymes exhibited a half-life of 53 min in the plasma after tail intravenous injection. Next, we studied the pharmacokinetic behavior of C-dot SOD nanozymes by tracking the fluorescent signal of the Cy5.5 labeled C-dots in MCAO mice (Supplementary Fig. 12). At 2 h post-injection, C-dot SOD nanozymes mainly accumulated in the liver and kidney with small amounts in the spleen and lung as well. The accumulation of C-dot SOD nanozymes in the liver dropped significantly 6 h after injection. Interestingly, the amount of C-dot SOD nanozymes accumulated in the brain of MCAO mice did not diminish significantly over the 24 h period post-injection (Supplementary Fig. 12). As shown by the brain section imaging, the fluorescence of the infarcted model was substantially greater at 24 h post-injection compared to the sham mouse brain, demonstrating that the C-dot SOD nanozymes accumulate preferentially in the infarcted brain regions (Fig. 6b). These results may be caused by ischemic stroke partially damaging the blood brain barrier. In addition, the ROS damage targeting ability of C-dot nanozymes also facilitated their accumulations in the damaged brain area.

We then evaluated the therapeutic effects of C-dot SOD nanozymes on the ischemic stroke male mice model. First, we established the optimal dose of C-dot SOD nanozymes for remedying MCAO mice. As shown in Fig. 6c and Supplementary Fig. 13, the MCAO mice administered with dose of 2.5 mg/kg C-dot SOD nanozymes exhibited the lowest infarcted area. Doses of C-dot SOD nanozymes less than 2.5 mg/kg may not have had sufficient concentration to effectively scavenge the superoxide radicals in vivo, thus unable to mitigate the ROS in ischemic stroke. On the other hand, doses higher than 2.5 mg/kg were thought to over-scavenge ROS in vivo, disrupting the redox balance of cells, which is critical to the survival and proliferation of the cells. Under the optimal concentration of 2.5 mg/kg, we investigated the therapeutic effects of three kinds of C-dot SOD nanozymes with specific activities of $1.6 \times 10^3$, $3.8 \times 10^3$ and $1.1 \times 10^4$ U/mg (Supplementary Table 2), representing low, medium, and high activity, on recovering following a stroke. The neuro score and infarct size were evaluated at 24 h after the ischemic-reperfusion. As shown in Fig. 6d and e, compared to PBS group, individual injection of C-dots-NaBH4-HNO3 and C-dots (with medium and high SOD-like activities, respectively) significantly reduced the infarcted area in the brain. C-dots-PS group, on the other hand, had a lower impact on the infarcted area of the brain, which is consistent with their low of SOD-like activity. The neurological scores after the treatment further indicate that C-dot nanozymes significantly alleviate the neurological and cognitive damage caused by ischemic stroke in vivo (Fig. 6f). The MCAO mice treated with PBS buffer showed evident behavioral disorders post-ischemic stroke with a neurological damage score of 3–4. In contrast, the MCAO mice treated with C-dot nanozymes with the highest SOD-like activity scored 1–2, while the MCAO mice treated with C-dots-PS and C-dots-NaBH4-HNO3 scored 1-3. TUNEL assay showed that the apoptotic cells in the brains of MCAO mice were remarkably reduced when treated by C-dot nanozymes (Fig. 6g). Malondialdehyde (MDA) assay was used to detect the level of lipid peroxidation in the brain homogenate (Fig. 6h). The results showed that the level of lipid peroxidation in the brains of MCAO mice decreased to the level of the sham mice after the treatment of the three kinds of C-dots, indicating that the relief of neurological impairment is attributable to the reduction of ROS by C-dot SOD nanozymes. Several proinflammatory cytokines, including tumor necrosis factor-α (TNF-α), interleukin 1β (IL-1β), and interleukin 6 (IL-6) are important in infarction progression and tissue injury. The reduction in these inflammatory markers is related to the successful treatment of the ischemic stroke. As shown in Fig. 6i–k, levels of inflammatory factors including TNF-α, IL-1β, and IL-6 in MACO mice significantly decreased after treatment of C-dot nanozymes with the highest activity while those of the groups treated with C-dots-PS and C-dots-NaBH4-HNO3 exhibited less or no significant changes. The above results indicate that C-dot SOD nanozymes successfully reduce ROS-mediated oxidative damage in ischemic stroke model, and a better therapeutic effect was observed in C-dots with higher SOD-like activity.

### In vivo toxicological analysis of C-dot SOD nanozyme

C-dots have been known to exhibit good biocompatibility. The in vivo toxicity of C-dot SOD nanozymes were systematically evaluated to ensure their safety. The safety evaluations were performed at the optimal therapeutic dose of 2.5 mg/kg. As shown in Fig. 7a, the weight

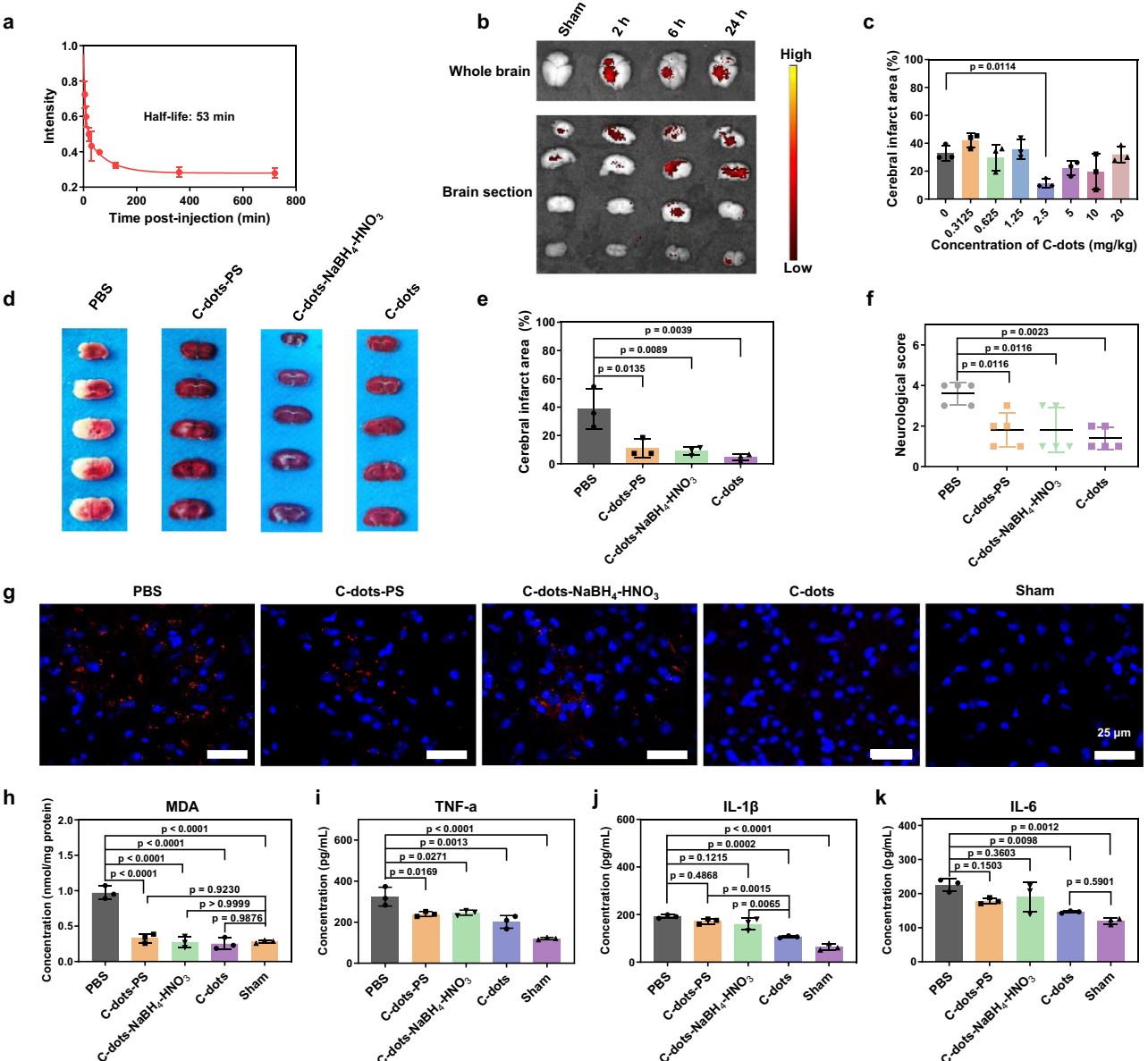

**Fig. 6 | Brain protection of C-dot SOD nanozymes against middle cerebral artery occlusion (MCAO)-induced ischemic and reperfusion injury. a** Half-life analysis of C-dot SOD nanozymes in plasma (*n* = 3 mice). **b** Ex vivo fluorescence imaging analyses of the accumulations of C-dot SOD nanozymes (labeled with Cy5.5) in the brains of sham (24 h post-injection) and MCAO mice (2 h, 6 h, and 24 h post-injection), and corresponding brain sections (*n* = 3 mice). **c** The cerebral infarcted area analyses of MCAO mice treated with different dosages of C-dot SOD nanozymes for 24 h (*n* = 3 mice). **d** Representative 2,3,5- triphenyltetrazolium chloride-stained brain sections and (**e**) quantification of cerebral infarct areas of MCAO mice treated with different C-dot nanozymes (*n* = 3 mice). **f** Neurological score analyses of the MCAO mice treated with different C-dot nanozymes for 24 h (*n* = 5 mice). **g** Representative images of TUNEL staining in the brain sections (*n* = 3 mice, scale bar = 25 μm), **h** Malondialdehyde (MDA) assay in the brain homogenate, and ELISA assay of inflammatory factors (**i**) TNF-α, (**j**) IL-1β and (**k**) IL-6 of the infarcted brain of MCAO mice treated by different C-dot nanozymes (*n* = 3 mice). *P* values are determined with one-way ANOVA Tukey's multiple comparisons test. Source data are provided as a Source Data file.

change trend of the mice injected with C-dot SOD nanozymes was consistent with that of the control group. No significant difference was observed between C-dot SOD nanozymes treated groups and the control groups in the levels of alanine transaminase (ALT), aspartate transaminase (AST), alkaline phosphatase (ALP), urea, and creatinine (CREA) of mice at 7 and 30 days post-injection (Fig. 7b), indicating that C-dot SOD nanozymes exhibit minimal effects on the kidney and liver of the mice. Moreover, the routine blood results of healthy mice at 7 days and 30 days with injection of C-dot SOD nanozymes were not significantly different from the PBS administered group (Fig. 7c). After injection of C-dot SOD nanozymes, the hematoxylin and eosin (H&E) staining major organs, including heart, liver, spleen, lung, kidney, and

brain showed no signs of damage, demonstrating the good biocompatibility of C-dot SOD nanozymes (Fig. 7d). These data show the satisfactory biocompatibility of C-dot SOD nanozymes at the therapeutic dose in healthy mice.

Overall, we have synthesized C-dot nanozyme with satisfactory SOD-like activity through the oxidation of activated charcoal. The functional groups, including carboxyl, hydroxyl, carbonyl groups, as well as C = C on the surface of C-dots were tuned by chemical reactions to investigate the nanozyme catalytic mechanism. Combining with theoretical calculation, we found that the hydroxyl and carboxyl groups act as the substrate-binding sites and the carbonyl groups coupled with π-system serve as the active catalytic sites. Hence,

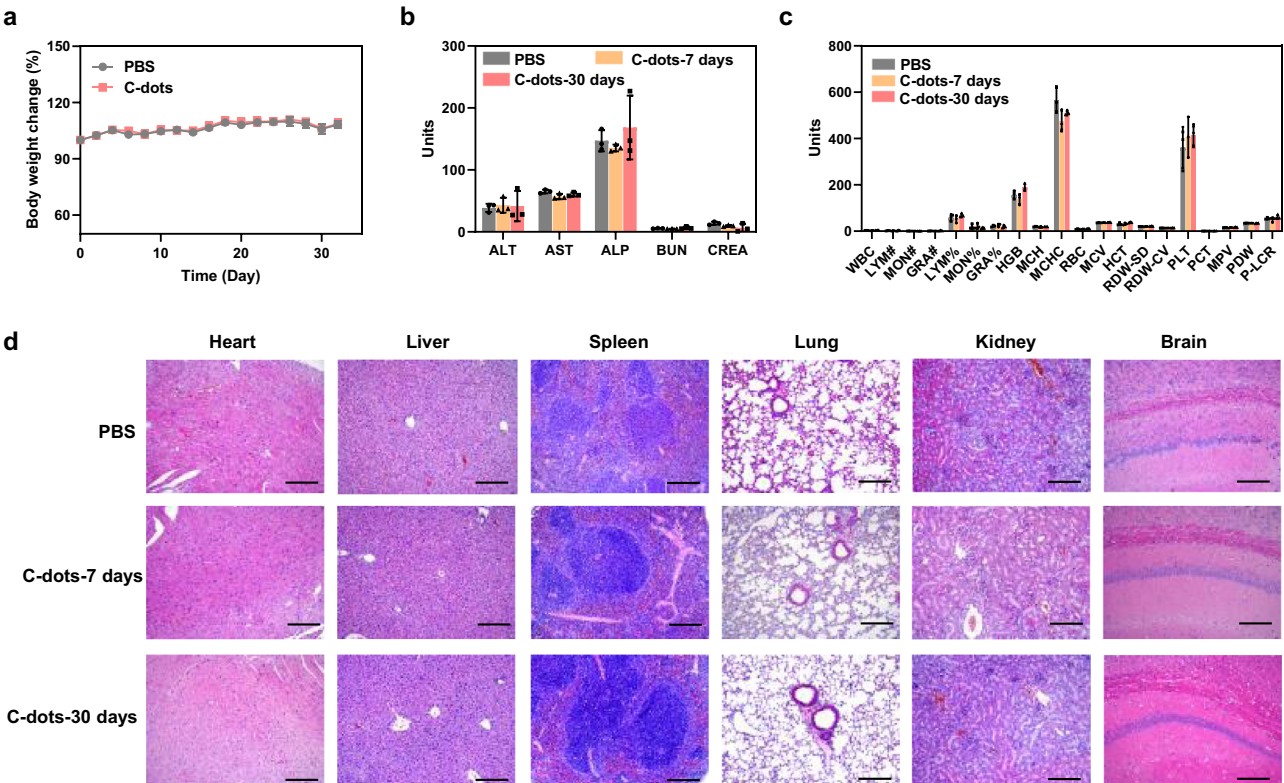

**Fig. 7 | In vivo biosafety analysis of C-dot SOD nanozymes. a** Body weight change in healthy mice after treatment with PBS and C-dot SOD nanozymes, respectively (*n* = 4 mice). **b** Blood biochemical analysis and **c** routine blood test of healthy mice at 7 days post-injection of C-dot nanozymes and 30 days post-injection of PBS and C-dot SOD nanozymes, respectively (n = 3 mice). The units of different parameters in serum biochemical analyses and blood routine tests are shown in source data. Source data are provided as a Source Data file. **d** Hematoxylin-eosin staining images of the tissue sections of heart, liver, spleen, lung, kidney, and brain obtained from healthy mice after treatment with PBS, and C-dot SOD nanozymes at 7 days and 30 days post injection; (*n* = 3 mice); scale bar = 200 μm.

increasing the quantity of hydroxyl, carboxyl, and carbonyl groups on the surface of C-dots with adequate π-system by chemical modifications might lead to significant SOD-like activity. Moreover, unlike many nanoparticles, which are harmful to biological systems, C-dot SOD nanozymes exhibit satisfactory biocompatibility and are capable of crossing cell membranes that are damaged by a high level of ROS. C-dot SOD nanozymes efficiently scavenged intracellular excess ROS to protect cells from oxidative damage. In the in vivo ischemic stroke model, C-dot nanozymes were able to specifically accumulate in the injured brain area, and scavenge ROS to mitigate the damage induced by the stroke. This work brings to light a SOD activity for C-dot nanozymes, identifies the mechanism of the SOD-like activity, and investigates the in vivo behavior of the C-dot SOD nanozymes both in terms of targeting and therapeutic effects. More importantly, the C-dot SOD nanozyme overcomes most drawbacks of natural enzymes including low stability, high cost, difficulty of preparation, and being burdensome for mass production, which makes them a promising substitute for natural SOD and great application potential in industrial, medical, and biological fields.

## Methods

### Ethical regulations

All research complied with all relevant ethical regulations. Animal studies were performed following the protocol approved by the Institutional Animal Care and Use Committee of the Institute of Biophysics, Chinese Academy of Sciences (SYXK2019021).

### Chemicals and materials

Activated charcoal, 1,3-propane sultone (PS), anhydrous acetonitrile, sodium bisulfite, hydroiodic acid (HI, 55–57%) and 5,5-dimethyl-1-pyrroline-N-oxide (DMPO) were purchased from Aladdin Chemical Reagent Co., Ltd. Carbon black, thionyl chloride (SOCl2), and 3,3′,5,5′-tetramethylbenzidine dihydrochloride (TMB) were purchased from Shanghai McLin Biochemical Technology Co., Ltd. Graphite powder, 1, 4-dioxane, and sodium borohydride (NaBH4) were purchased from Sinopharm Chemical Reagent Co., Ltd. Triethylamine (TEA) was purchased from Energy Chemical Co. Ltd. Sodium chloride(NaCl), sodium hydroxide(NaOH), and sodium bicarbonate (NaHCO3) were purchased from Tianli Enterprise Group Co., Ltd. Sulphuric acid (H2SO4, 98%), and nitric acid (HNO3, 65–68%) were purchased from local supplies. Acetonitrile was purchased from Tianjin Kermel Chemical Reagent Co., Ltd. Sulfo-Cyanine5.5 amine was obtained from Lumiprobe. Superoxide dismutase assay kit (S311) was purchased from Dojindo Molecular Technologies Co., Ltd. Superoxide dismutase was provided by Siyomicro BIO-TECH Co., Ltd. Mito-Tracker Green, total superoxide dismutase assay kit with WST-8, 2′,7′-dichlorodihydrofluorescein diacetate (DCFH-DA), and dihydroethidium (DHE) were purchased from Beyotime Chemical Reagent Co., Ltd. 4′, 6-diamidino-2- phenylindole (DAPI) was obtained from Roche Applied Science. Recombinant anti-LAMP1 antibody was purchased from Abcam. 2,3,5-Triphenyltetrazolium chloride (TTC, DK0005) was purchased from Leagene Biotechnology. Malondialdehyde (MDA, A003-2) was purchased from Nanjing Jiancheng Bioengineering Institute. Interleukin 1β (IL-1β) Detection Kit (MDL, MD6758), Interleukin 6 (IL-6) Detection Kit (MDL, MD123475) and Tumor necrosis factor α (TNF-α) Detection Kit (MDL, MD7125) were purchased from Medical Discovery Leader. All chemical reagents were not further purified and were used directly, and detection kits were used according to instructions of manufacturers. All aqueous solutions used in this work were prepared with deionized water with a resistivity of 18.2 MΩ·cm.

## Instrumentation

Transmission electron microscopy (TEM) images were obtained by using a FEI Tecnai G2 F30 (FEI, USA) at an acceleration voltage of 300 kV. Powder X-ray diffraction (XRD) data were collected by using a Bruker D8 ADVANCE (Germany) with a scan rate of 6 ° / min. FT-IR spectra were recorded by a Thermo Fisher Nicolet 5700 (USA). Raman spectra were performed by a Thermo Fisher DXR2xi Raman Imaging Microscope (USA) under excitation wavelength of 532 nm. The proton magnetic resonance ($^1$H NMR) spectra were recorded using an AVANCE III HD (USA) spectrometer (600 MHz, $D_2O$ as solvent). X-ray photoelectron spectroscopy (XPS) spectra were recorded by a Thermo Escalab 250Xi (USA). Electron spin resonance (ESR) spectra were recorded by a Bruker A300-9.5/12 (Switzerland) at room temperature. Confocal laser scanning microscopy images were obtained by Olympus FluoView FV-1000 (Japan). Flow cytometry data were collected by FACS Calibur™, Becton Dickinson (USA). Microplate absorbance was measured using Tecan Spark 20 M multi-mode microplate reader, Switzerland. The fluorescence imaging was done using an in vivo imaging instrument (IVIS Lumina 3, PE, USA), and images were obtained by IVIS Living Image 3.0 software (PerkinElmer, USA). Leica DM3000 microscope (Leica, Wetzlar, Germany) was used to image tissue sections. All the calculations were carried out using the Gaussian 09 package (Gaussian, Inc., Wallingford CT).

## Synthesis of C-dot SOD nanozymes

In total, 0.5 g of bulk carbon material (activated charcoal, carbon black, or graphite powder) was added to 50 mL of $HNO_3$ and $H_2SO_4$ ($V_{HNO_3} : V_{H_2SO_4} = 1 : 1$) and refluxed for a given time. The corresponding solution was neutralized with $NaHCO_3$. The resultant solution containing C-dots was purified by filtering (0.22 μm membrane filters) and dialyzing for ca. a week. Then the C-dots solution was condensed and ultra-filtered by using a Millipore centrifuge filter device with a molecular-weight cut-off (MWCO) membrane of 100 kDa. The separated fraction with weight equivalent to <100 kDa was collected and stored for use. It is noted that the optimal time of C-dots prepared from graphite powder, carbon black, and activated charcoal with the highest enzymatic activity were 10 h, 1 h and 1.5 h, respectively.

## Synthesis of C-dots-PS

In total, 1 mL of C-dots (5 mg/mL in water) and 1,3-propanesultone (PS) (1 g) were added to 10 mL of 1, 4-dioxane solution, then 1 mL of TEA was added to this mixture. The reaction mixture was stirred for 24 h at 40 °C. After then, the solvent of the mixture was removed by rotary evaporation and the resulting product was redispersed in water and dialyzed in a 3500-Da dialysis bag for four days. 0.1 M NaCl solution was used to remove TEA salt through dialysis for the first day, and then ultrapure water was used for removing NaCl and other impurities for the other three days[56].

## Synthesis of C-dots-PS-HCl

C-dots-PS (5 mg) was added to HCl solution (0.1 M, 10 mL) and then refluxed for 12 h. The resulting solution was neutralized with $NaHCO_3$ and then dialyzed for 3 d.

## Synthesis of C-dots-NaBH₄

C-dots (20 mg) was added to $NaBH_4$ solution (0.5 M, 50 mL), the mixture was stirred for 24 h at room temperature. The resulting product was neutralized with HCl and further dialyzed for 3 d[58].

## Synthesis of C-dots-NaBH₄-HNO₃

C-dots-NaBH$_4$ (1 mg) was added to $HNO_3$ solution (5 M, 10 mL), then the mixture was stirred for 36 h at 40 °C. The resulting solution was neutralized with $NaHCO_3$ and further dialyzed for 3 d.

## Synthesis of C-dots-NaOH-200 °C

C-dots (20 mg) was added to NaOH solution (5 M, 50 mL), the mixture was transferred to a Teflon-lined autoclave and heated for 24 h at 200 °C. The resulting solution was neutralized with HCl, and then dialyzed for 3 d[58].

## Synthesis of C-dots-NaOH-HI

In total, 10 mg of C-dots-NaOH-200 °C was added to 20 mL acetic acid containing 2 mL of HI (55–58%). After refluxed for 24 h, the solution was immediately poured into 50 mL of sodium bisulfite solution (4%), and then dialyzed for 3 d to remove impurities[38].

## Synthesis of C-dots-NaOH-40 °C

C-dots (5 mg) was added to NaOH solution (0.5 M, 10 mL) and stirred for 24 h at 40 °C. The resulting solution was neutralized with HCl and then dialyzed for 3 d.

## The SOD-like activity of C-dots

The SOD-like activities of the C-dots were tested by using a Total Superoxide Dismutase Assay Kit (S311-10, Dojindo Molecular Technologies) according to the instructions of manufacturer. The SOD-like activity of C-dots with a series of concentrations was represent as the inhibition rate of the competitive WST-1 reaction.

## Mitochondria/ lysosome localization

SH-SY5Y cell line was purchased form Pricella, CRL-2266. Briefly, SH-SY5Y cells were first seeded and cultured for 12 h on a 6-well plate. After removing the cell medium, adherent cells were treated with C-dots (25 μg/mL) for 8 h. To eliminate the excess nanoparticles, cells were washed three times with PBS. To identify the mitochondria, the cells were incubated with 1 mL of PBS containing 100 nM Mitotracker for an additional 30 min. To identify lysosomes, cells were fixed for 5 min in 4% formaldehyde in PBS before being permeabilized with 0.1% Triton X-100. After three washes with PBS, the cells were blocked in 5% goat serum at 37 °C for 30 min before incubation with an anti-Lamp1 mAb (1:200, Abcam) for 1 h at 37 °C. The cells were then washed three times before being treated with Alexa Fluor 488 goat anti-rabbit secondary antibody (1:500; ThermoFisher). 4', 6'-diamidino-2-phenylidole (DAPI, 1 μg/mL, Roche Applied Science) was used to stain the nuclei of the cells. Finally, the cells were washed with PBS three times. Confocal laser scanning microscopy (Olympus FluoView FV-1000, Tokyo, Japan) were used to obtain the cell images.

## Accumulation of C-dots in SH-SY5Y cells

Briefly, SH-SY5Y cells were plated in six-well plates at $1.5 \times 10^5$ cells / well and allowed to settle overnight for adherence. C-dots-Cy5.5 (25 μg/mL) were then added into wells and incubated for 8 h of incubation in the presence of $H_2O_2$ (50 μM, 100 μM, and 200 μM). Flow cytometry was used to assess the fluorescence intensities of the cells, which were then analyzed using the FlowJo 7.6 software.

## Intracellular ROS/ O₂•⁻ scavenging in SH-SY5Y cells

The fluorescent probe 2',7'-dichlorofluorescein diacetate (DCFH-DA), and Dihydroethidium (DHE) were employed to evaluate the generation of intracellular ROS and $O_2^{\cdot-}$, respectively. SH-SY5Y cells were plated and allowed to settle overnight for adherence in six-well plates. C-dots (10 μg/mL) were then added into wells and incubated in the presence of paraquat (250 μM) for 24 h. The cells were then incubated with 1 mL of PBS containing DCFH-DA (10 μM) or DHE (10 μM) for 45 min. Finally, flow cytometry was used to investigate the fluorescence intensities which were then analyzed using FlowJo 7.6 software.

The cell viability of SH-SY5Y cells co-incubated with C-dots and paraquat was measured using a Cell Counting Kit-8 (CCK-8) from Dojindo Chemical Technology (Beijing) Co., Ltd. In brief, SH-SY5Y cells were plated in 96-well plates at $10^4$ cells/well and allowed to settle

overnight for adherence before adding paraquat (final concentration of 250 μM) and various concentrations of C-dots (final concentrations of 0, 2.5, 10, 20 μg/mL) to the wells. The assay was performed in accordance to the manufacturer's instructions, and the absorbance at 450 nm was quantified by a microplate reader.

### Animal model
C57BL/6 J mice (male, 8–10 weeks) were group-housed 5 mice per cage in temperature (20–26 °C) and humidity (40%–70%) housing rooms on a 12 h light, 12 h dark cycle. The focal cerebral ischemia-reperfusion model was constructed based on the reversible middle cerebral artery occlusion (MCAO) method proposed by Longa et al.[66], with modification. Briefly, after a median carotid incision the common carotid artery (CCA), internal carotid artery (ICA), and external carotid artery (ECA) of the mice were isolated from the surrounding tissues. To occlude the middle cerebral artery (MCA), a silicone-wrapped suture was inserted into the ICA for 10 mm. The end of the suture was secured to the skin, and the wound was cleaned and sutured. After 2 h of occlusion, the suture was gently removed to allow for reperfusion. As a control, sham surgery was employed, which all arteries are exposed throughout surgical period but the suture is not inserted into the MCA.

### Pharmacokinetics analysis
C-dots-Cy5.5 (2.5 mg/kg) were injected intravenously into the tail vein of C57BL/6 mice to analyze the pharmacokinetics of C-dots ($n = 3$ mice). At different time points (5 min, 10 min, 20 min, 1 h, 2 h, 3 h, 6 h, 12 h) after injection of C-dots-Cy5.5, 10 μL of blood was drawn from the tail vein and combined with 5 μL of sodium citrate anticoagulant. To separate the supernatant, the blood sample mixtures were centrifuged at $1000 \times g$ for 15 min. 10 μL of the supernatant were added to 490 μL of PBS then centrifuge at $12,000 \times g$ for 10 min before pipetting 100 microliters of supernatant in a 96-well black plate. A microplate reader was used to measure the fluorescence intensity in the sample using an excitation wavelength of 675 nm, and emission wavelength of 695 nm. The in vivo circulating half-life of C-dot SOD nanozyme in blood stream is calculated by a two-phase decay model by GraphPad 8.

### In vivo tracking of C-dots
After 2 h of occlusion-reperfusion, C-dots-Cy5.5 (2.5 mg/kg) was injected intravenously. After 2, 6, 24 h of injection ($n = 3$ mice per group), the mice were sacrificed and their brain, heart, liver, spleen, lung, kidney were collected, radiance signals in each organ were measured using IVIS Spectrum Imaging System (Xenogen). For brain sections, the brain of sham mice 24 h post injection of C-dots-Cy5.5 and 2, 6, 24 h was cut 2.0 mm thick to a total of 4 pieces and measured using IVIS Spectrum Imaging System (Xenogen). The concentrations of Cy5.5 in each tissue were determined by using IVIS Spectrum Imaging analysis software.

### In vivo neuroprotection evaluation
For the in vivo anti-ischemic stroke efficacy testing, mice were randomly divided into four groups, normal saline group and the C-dot treated groups, the dosages of three kinds of C-dot SOD nanozymes with specific activities of $1.6 \times 10^3$, $3.8 \times 10^3$ and $1.1 \times 10^4$ U/mg are 2.5 mg/kg. The treated mice were anesthetized and sacrificed after 24 h treatment. Brains were immediately removed and used for 2,3,5-triphenyltetrazolium chloride (TTC) staining ($n = 3$ per group). Briefly, the brains were frozen at −20 °C for 30 min then cut into five 1.5-mm thick sections and stained with 2% TTC solution in 37 °C for 20 min before adding to fixing solution. The infarct area was quantified with Image J.

Neurological scores were evaluated by observing the motor ability of mice in each group in a blinded fashion to avoid bias ($n = 5$ mice per group). The higher the score, the more severe the neurological injury. In detail, normal behavior, no neurological damage: 0 points; left front paw unable to fully extend, mild neurological damage: 1 point; moves in a circle to the left (paralyzed side), moderate neurological damage: 2 points; body severely falls to the left side (paralyzed side), severe neurological damage: 3 points; unable to walk, loss of consciousness: 4 points.

For TUNEL staining, the treated mice were anesthetized and sacrificed after 24 h treatment. Brains were immediately removed and further fixed with 4% paraformaldehyde for 24 h, followed by dehydration with 15 and 30% sucrose stepwise at 4% overnight. In total, 20 μm sections were prepared. Cell apoptosis in ischemic penumbra was stained with One-Step TUNEL Apoptosis Assay Kit according to the manufacture's instruction. MDA assays were performed on brain tissue homogenates using an MDA assay kit (Beyotime, China).

For ELISA assay, whole blood samples ($n = 3$ mice per group) were collected in the serum separation tube at room temperature for 2 h, then centrifuge at $1000 \times g$ for 20 min, take the supernatant, and place the supernatant at −20 °C. Interleukin 1β (IL-1β) Detection Kit (MDL, MD6758), Interleukin 6 (IL-6) Detection Kit (MDL, MD123475) and Tumor necrosis factor α (TNF-α) Detection Kit (MDL, MD7125) were used in this study. The OD value was measured at a wavelength of 450 nm.

### In vivo safety evaluation
All mice were randomly assigned to one of two groups: test group or control groups. The mice in the test group were given an intravenous injection of C-dots (2.5 mg/kg) whereas the mice in control group were injected with PBS. Body weight was measured every other day following injection ($n = 4$ mice per group). Major organ tissues (brain, heart, liver, kidney, spleen, and lung) and blood samples were collected at 7 and 30 days of injection and blood biochemical analysis was performed ($n = 3$ per group). Major organs were immediately fixed in 4% paraformaldehyde, then embedded in paraffin and sectioned into 5 μm slices, before being stained with hematoxylin-eosin.

### Statistical analysis
General statistical data were analyzed by Image J, Origin 8, Nano Measurer 1.2, MestReNova 5.3.1-4696, XPSPEAK41, DigitalMicrograph 3.7.4, ZEN 2010, FlowJo 7.6.1 and Graphpad prism 8. One-way ANOVA Tukey's multiple comparisons test was used to determine statistical significance by GraphPad Prism 8.0 (GraphPad Software, Inc.).

### Reporting summary
Further information on research design is available in the Nature Portfolio Reporting Summary linked to this article.

## Data availability
Data supporting the findings of this work are available within the paper and its Supplementary Information files. Source data are provided with this paper and the raw data are available upon request to the corresponding authors. Source data are provided with this paper.

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

## Acknowledgements

This work was supported by the National Natural Science Foundation of China (82122037 (K.F.), 81930050 (X.Y.), 32171392 (C.L.), 82000523 (M.Z.), 82101411 (J.H.), and 21805021 (C.L.)), National Key Research and Development Program of China (2021YFC2102900 (K.F.)), Youth Innovation Promotion Association of Chinese Academy of Sciences (2019093 (K.F.)), CAS Interdisciplinary Innovation Team (JCTD-2020-08 (K.F.)), the Natural Science Foundation of Shaanxi Province of China (2021JQ-009 (C.L.)), China Postdoctoral Science Foundation (2020M683449 (C.L.)), the "Young Talent Support Plan" of Xi'an Jiaotong University, China (YX6J001 (M.Z.)). We thank Dr. Zijun Ren and Dr. Gang Chang at the Instrument Analysis Center of Xi'an Jiaotong University for assisting with TEM analysis and [1]H-NMR analysis, respectively. We also thank Dr. Zhuoran Wang (Institute of Biophysics, Chinese Academy of Sciences) for technical support, and Mr. Haolin Cao (Institute of Biophysics, Chinese Academy of Sciences) for preparing Chemdraw figures.

## Author contributions

K.F. and C.L. conceived the project. W.G., J.H., C.L., M.Z., and K.F. designed, conducted the experiments and wrote the manuscript. L.C., X.G. provided theoretical calculation and wrote the relevant content. X.M., Y.M., L.C., X.G. and K.T. provided relevant experimental technical support and helped with the investigations. C.L., M.Z., K.F., D.P., and X.Y. supervised this study and revised the manuscript. All authors critically revised the article and approved the final manuscript.

## Competing interests

The authors declare no competing interests.
