## [Peer Review File · Nature Communications]

Deciphering the catalytic mechanism of superoxide dismutase activity of carbon dot nanozymeREVIEWER COMMENTS

Reviewer #1 (Remarks to the Author):

The authors designed a carbon nanodot (C-dot) SOD nanozyme with ultra-high catalytic activity and investigated the surface state-related catalytic mechanism of highly active C-dot SOD nanozymes, as well as the potential to protect neuron cells during ischemic stroke. Impressively, the SOD-like activity of C-dots is at least several-fold higher than natural SODs and the reported SOD nanozymes under the same test conditions. Meanwhile, the surface structure tuning and theoretical calculations deeply revealed the hydroxyl and carboxyl groups act as the substrate-binding sites and the carbonyl groups coupled with π -system serve as the active catalytic sites of C-dot SOD nanozyme. In the ischemic stroke model of mouse, C-dot SOD nanozyme exhibited the capability to overcome the blood-brain barrier and reduce the intracellular ROS, showing neuron protection effect. This work would be of great significance to support the rational design of C-dot SOD nanozymes with desirable catalytic activity, and to promote the practical application of SOD nanozymes. Therefore, I would like to recommend this work be accepted for publication in Nature Communications after the following several issues are addressed.

1. In this manuscript, the authors designed the C-dots with ultra-high SOD-like activity (C-dot SOD nanozyme, >10000 U/mg) and revealed their catalytic mechanism. Many nanozymes exhibit multienzyme-like activities, how about the C-dots? Whether they have other enzyme-like activities such as POD-, OXD- or CAT-like activities.
2. In Scheme 1, in order to understand the manuscript more intuitively, the Schematic illustration should be labeled (a, b) for the convenience of readers.
3. In Figure 3c, there are two abbreviations for C-dots treated in NaOH solution at 40°C ("C-dots-NaOH (40°C)" and "C-dots-NaOH-40°C"), the author should revise this.
4. In Figure 4, in order to be consistent with other figures, it is recommended to mark the labels (a, b, c, d) from left to right and from top to bottom.
5. In Figures 5d and 5f: To correctly study co-localization between C-dots and mitochondria or Lysosome, a Pearson's coefficient or other correlation studies should be provided.
6. In this manuscript, the authors claimed that C-dots could be accumulated in the mitochondria of cells. Why possess the oxidation damage targeting? An explanation may be needed.
7. In Figure 7d, it is recommended to add the scale bar consistent with other figures.
8. In the supplementary "Animal model", the sex and age of the laboratory animal should be indicated.
9. In the supplementary "Animal model", the authors claimed that the construction of the focal cerebral ischemia-reperfusion model was based on the reversible middle cerebral artery occlusion (MCAO) method proposed by Longa et al., with modification. The related reference should be cited.
10. The "Results and discussions" are supposed to be divided into separate sections with headings, which should be used as a reading guideline.
11. On page 28, the "5." in the title of "5. Conclusions", should be removed from here, because it does not appear in the previous title.
12. On page 29, the "H-NMR" in the "Acknowledgements" should be changed to "1H-NMR".
13. For readers to better understand the nanozyme activity of C-dots, some related publications should be cited, such as Journal of Colloid and Interface Science 611, 545-553 (2022), ACS Nano 16, 9228-9239(2022), and Nano Today 45, 101530(2022).
14. In supplementary "References", the reference style is incorrect, which should be modified as per the guidelines.

Reviewer #2 (Remarks to the Author):

Specific Comments:

1. The present abstract highlights the results. Normally an abstract should include the above along with briefly stating the purpose of the study undertaken and meaningful conclusions based on the obtained results. I would expect a brief yet concise quantitative description in the abstract.
2. L27 - SOD nanozymes reported thus far typically exhibit low catalytic activity. How do authors

- testify and justify this issue in this manuscript? Any results data?
3. L29 - deeply investigated, the word "deeply" reads unnecessary.
 4. L50 - catalysts in biomedical applications. Enlist some with key examples and supporting references.
 5. L55 – Avoid redundant expressions, e.g., broad variety. To somehow, broad and variety present the same meaning in this sentence.
 6. L61-62 – Referencing is not right. For instance, Up to now, nanozyme research mostly revolves around oxidoreductase activities, including oxidase (OXD), peroxidase (POD), catalase (CAT), and superoxide dismutase (SOD). This sentence is supported later with Ref. 6,7. None of them coat all the above-mentioned enzymes.
 7. L67 – Avoid vagueness. a series of nanozymes. Enlist some with key examples and supporting references.
 8. L105 - Scheme 1. What is the basis of this scheme? Any analytical proof that confirms the given mechanism?
 9. The work is of primary importance with poor sampling size. The sample variability and reliability are low. More than half of the work is based on a single sample characterization. For example, the data given in the supplementary files lacks graphite powder, carbon black, and activated charcoal. Reconsider all results data and make sure to present the analysis of all samples in a consistent manner.
 10. What were the control samples? Any data for positive control/negative control?
 11. Figure 1 (d-f) can be merged as a single image.
 12. Table 1 is superficial, and many cells are empty. Authors cannot define superiority/inferiority with this little data. Move this into supplementary data and extend this through comparison.
 13. Figure 2 - (d) FT-IR, what is a point in highlighting the overexpressed broad peak in the range 3200-3500 and 2750-2250? what authors can explain the main peaks between 2000 and 1000 wavenumbers.? Major peak numbers should be assigned with representative functional groups.
 14. Results – Data are not statistically shown; thus, any results could not be interpreted well. The figures and tables not showing the statistical differences. The author should perform the statistical analysis by Tukey or Duncan test and indicate the significance in the superscript (a, b, c or in asterisk, *, **, ***) of each value.

General Comments:

Your manuscript has FIVE corresponding authors. Multiple corresponding authors are not encouraged. Please provide a detailed explanation about why this manuscript needs FIVE corresponding authors. Please kindly note only the authors who have solid contributions to this manuscript can be listed as corresponding authors.

Responses to the reviewers' comments:

Note: The reviewers' comments are in black, and our Response are highlighted in blue.

Reviewer #1 (Remarks to the Author):

The authors designed a carbon nanodot (C-dot) SOD nanozyme with ultra-high catalytic activity and investigated the surface state-related catalytic mechanism of highly active C-dot SOD nanozymes, as well as the potential to protect neuron cells during ischemic stroke. Impressively, the SOD-like activity of C-dots is at least several-fold higher than natural SODs and the reported SOD nanozymes under the same test conditions. Meanwhile, the surface structure tuning and theoretical calculations deeply revealed the hydroxyl and carboxyl groups act as the substrate-binding sites and the carbonyl groups coupled with π -system serve as the active catalytic sites of C-dot SOD nanozyme. In the ischemic stroke model of mouse, C-dot SOD nanozyme exhibited the capability to overcome the blood-brain barrier and reduce the intracellular ROS, showing neuron protection effect. This work would be of great significance to support the rational design of C-dot SOD nanozymes with desirable catalytic activity, and to promote the practical application of SOD nanozymes. Therefore, I would like to recommend this work be accepted for publication in Nature Communications after the following several issues are addressed.

Reply: We really appreciate the reviewer's understanding of the merits of the present study and the positive comments.

1. In this manuscript, the authors designed the C-dots with ultra-high SOD-like activity (C-dot SOD nanozyme, >10000 U/mg) and revealed their catalytic mechanism. Many nanozymes exhibit multienzyme-like activities, how about the C-dots? Whether they have other enzyme-like activities such as POD-, OXD- or CAT-like activities.

Reply: Thanks for your suggestion. According to the advice, we tested whether C-dots possess POD-, OXD- or CAT-like activities. The catalase-like activity was measured by monitoring the decomposition of H_2O_2 and the generation of oxygen, while the peroxidase- and oxidase-like activities were detected by monitoring the oxidization of 3,3',5,5'-tetramethyl-benzidine (TMB). We have added the results to the revised

version as **Supplementary Fig. 6** and discussion in the manuscript (Page 9) as follows.

In addition to SOD-like activity, we also tested whether C-dots possessed other enzyme-like activities, such as catalase-, peroxidase- and oxidase-like activities. The catalase-like activity was measured by monitoring the decomposition of H_2O_2 . In contrast, the peroxidase- and oxidase-like activities were detected by monitoring the oxidization of 3,3',5,5'-tetramethyl-benzidine (TMB) in the presence of H_2O_2 and dissolved oxygen, respectively. As shown in Supplementary Fig. 6, no significant catalase, peroxidase, or oxidase-like activities were detected in the C-dots. Therefore, the relatively exclusive SOD-like behavior of as-prepared C-dots allows for a more precise investigation into the mechanism of their catalytic activity.

Supplementary Fig. 6. Detection of CAT-like activity of C-dots by monitoring the elimination of H_2O_2 (a), and the generation of oxygen (b), detection of POD (c), and OXD-like (d) activities of C-dots.

2. In Scheme 1, in order to understand the manuscript more intuitively, the Schematic illustration should be labeled (a, b) for the convenience of readers.

Reply: Thanks for your kind advice. The corresponding correction has been added in the revised manuscript.

3. In Figure 3c, there are two abbreviations for C-dots treated in NaOH solution at 40°C (“C-dots-NaOH (40°C)” and “C-dots-NaOH-40°C”), the author should revise this.

Reply: The corresponding correction has been done in the revised manuscript.

4. In Figure 4, in order to be consistent with other figures, it is recommended to mark the labels (a, b, c, d) from left to right and from top to bottom.

Reply: Thank you for your helpful suggestion. The **Fig. 4** has been revised as follows.

Revised Fig. 4 Proposed SOD-like activity of C-dot nanozyme with and without hydroxyl groups. **a** Proposed reaction pathway to achieve the SOD-like catalytic cycle of C-dot nanozyme without hydroxyl groups. **b** Proposed reaction pathway to achieve the SOD-like catalytic cycle of C-dot nanozyme with hydroxyl groups. **c** Gibbs free energy profile for a SOD-like catalytic cycle of C-dot nanozyme without hydroxyl groups. **d** Gibbs free energy profile for a SOD-like catalytic cycle of C-dot nanozyme with hydroxyl groups.

5. In Figures 5d and 5f: To correctly study co-localization between C-dots and mitochondria or Lysosome, a Pearson's coefficient or other correlation studies should be provided.

Reply: As suggested, we analyzed the Pearson's co-localization coefficient of C-dots and mitochondria or lysosome in **Figs. 5d** and **f** by using Image J. The discussion has been added in the revised version (Page 20) as follows.

Following the staining of mitochondria with Mitotracker, we found that C-dot SOD nanozymes showed mitochondrial accumulation (Fig. 5d) with a Pearson's correlation coefficient of 0.43⁶⁷. Furthermore, co-staining C-dots with lysosome showed no visible overlap (Fig. 5f) with a Pearson's correlation coefficient of 0.03, indicating no accumulation of C-dot SOD nanozyme in the lysosome. These results demonstrated that the C-dot SOD nanozymes are capable of targeting the mitochondria by overcoming the cell membrane.

6. In this manuscript, the authors claimed that C-dots could be accumulated in the mitochondria of cells. Why possess the oxidation damage targeting? An explanation may be needed.

Reply: We thank the reviewer's interesting question. Previous studies indicated that graphene -based nanomaterials could cross the cell membrane in tumor cells with high ROS characteristics (*Adv. Mater.* 31, 1807456 (2019)). In this manuscript, to assess whether the C-dot SOD nanozyme possesses the oxidation damage targeting ability, we incubated C-dots-Cy5.5 with SH-SY5Y cells treated with different concentrations of H₂O₂. As shown in **Fig. 5b, c**, compared to normal cells, the higher concentration of H₂O₂, the more C-dots-Cy5.5 was accumulated in cells. The results demonstrated that the targeting ability of C-dot SOD nanozyme to oxidation-damaged cells might be due to the better permeability of membrane damaged by oxidative stress, which was consistent with the previous report that graphene-based nanoparticles target tumor cells by a cell membrane permeability targeting mechanism⁶⁶.

We have added the discussion in the revised manuscript as follows.

As shown in the confocal images (Fig. 5b), the higher concentration of H₂O₂, the

more C-dot SOD nanozymes accumulated in cells. The accumulation of C-dots in the cells was quantified by using flow cytometry (Fig. 5c). As shown in Fig. 5e, C-dots accumulated significantly more in 200 μM H_2O_2 treated cell than PBS treated cells. These results indicate the targeting ability of C-dot SOD nanozymes to oxidation-damaged cells may due to the better permeability of membrane damaged by oxidative stress, which is consistent with the previous report that graphene-based nanoparticles target tumor cells by a cell membrane permeability targeting mechanism⁶⁶.

7. In Figure 7d, it is recommended to add the scale bar consistent with other figures.

Reply: Thank you for your helpful suggestion. We have added scale bars to all the figures in the revised manuscript as follows.

Revised Fig.7d. Hematoxylin-eosin staining images of the tissue sections of brain, heart, liver, spleen, lung, and kidney obtained from healthy mice at 7 days and 30 days post injection; scale bar = 200 μm .

8. In the supplementary “Animal model”, the sex and age of the laboratory animal should be indicated.

Reply: As suggested, we have added the sex and age of the laboratory animal in the revised supplementary information as follows.

Briefly, C57BL/6J mice (male, 6-8 weeks) were intraperitoneally anesthetized with 10% chloral hydrate at a dose of 0.3 mL/kg. Then, the mice were sterilized in the neck, and the common carotid artery (CCA), internal carotid artery (ICA), and external carotid artery (ECA) were isolated from the surrounding tissues after a median carotid incision.

9. In the supplementary “Animal model”, the authors claimed that the construction of the focal cerebral ischemia-reperfusion model was based on the reversible middle cerebral artery occlusion (MCAO) method proposed by Longa et al., with modification. The related reference should be cited.

Reply: Thanks for the suggestion. The relevant reference was added as follows.

The construction of the focal cerebral ischemia-reperfusion model was based on the reversible middle cerebral artery occlusion (MCAO) method proposed by Longa et al.⁴, with modification.

10. The “Results and discussions” are supposed to be divided into separate sections with headings, which should be used as a reading guideline.

Reply: As suggested, we have divided the “Results and discussions” section into five parts:

- (1) Preparation and characterization of C-dot SOD nanozymes
- (2) Deciphering the mechanism of C-dot SOD-like activity
- (3) C-dot SOD nanozymes specifically targeting the ROS-damaged cells
- (4) C-dot SOD nanozymes alleviating neurological damage induced by ischemic stroke
- (5) *In vivo* toxicological analysis of C-dot SOD nanozyme

11. On page 28, the “5.” in the title of “5. Conclusions”, should be removed from here, because it does not appear in the previous title.

Reply: We have deleted the “5.” in the revised manuscript.

12. On page 29, the “H-NMR” in the “Acknowledgements” should be changed to “¹H-NMR”.

Reply: We have corrected this error in the revised manuscript as follows.

We also thank Dr. Zijun Ren and Dr. Gang Chang at the Instrument Analysis Center of Xi'an Jiaotong University for assisting with TEM analysis and ¹H-NMR analysis, respectively.

13. For readers to better understand the nanozyme activity of C-dots, some related publications should be cited, such as *Journal of Colloid and Interface Science* **611**, 545-

553 (2022), *ACS Nano* **16**, 9228–9239(2022), and *Nano Today* **45**, 101530(2022).

Reply: As suggested, we added the corresponding publications in the revised manuscript as *refs.* 51, 52, and 53.

14. In supplementary “References”, the reference style is incorrect, which should be modified as per the guidelines.

Reply: Thanks for your kind reminder. We have edited references in the revised supplementary information according to the guidelines.

Reviewer #2 (Remarks to the Author):

Specific Comments:

1. The present abstract highlights the results. Normally an abstract should include the above along with briefly stating the purpose of the study undertaken and meaningful conclusions based on the obtained results. I would expect a brief yet concise quantitative description in the abstract.

Reply: Thank you for your kind suggestion. We have revised the abstract as follows.

Nanozymes with superoxide dismutase (SOD)-like activity have attracted increasing interest due to their ability to scavenge superoxide anion, the origin of most reactive oxygen species in vivo. However, SOD nanozymes reported thus far have yet to approach the activity of natural enzymes. Here, we report a carbon dot (C-dot) SOD nanozyme with an unprecedented catalytic activity of over 10000 U/mg, comparable to that of natural enzymes. Through selected chemical modifications and theoretical calculations, we show that the SOD-like activity of C-dots relies on the hydroxyl and carboxyl groups for binding superoxide anions and the carbonyl groups conjugated with the π -system for electron transfer. Moreover, C-dot SOD nanozymes exhibit intrinsic oxidation-damaged cell targeting ability and significantly protect neuron cells in the ischemic stroke male mice model. Together, our study sheds light on the structure-activity relationship of C-dot SOD nanozymes, and demonstrates their potential for the treatment of oxidation stress related diseases.

2. L27 - SOD nanozymes reported thus far typically exhibit low catalytic activity. How do authors testify and justify this issue in this manuscript? Any results data?

Reply: We thank the reviewer for this interesting question. When preparing this manuscript, we comprehensively investigated the previous literatures on SOD nanozymes, and compared them with natural SOD. As shown in **Table S1**, compared with natural SOD, the reported SOD nanozymes exhibited much lower catalytic performance with specific activity ranging from 5 to 1300 U/mg. The low catalytic efficiency of SOD nanozymes is partly because the catalytic mechanism remains unexplored, resulting in difficulty in improving the catalytic performance. The C-dot SOD nanozyme in this work exhibited unprecedentedly high SOD-like activity of more than 10000 U/mg, which is even 2-fold better than the natural SOD we tested, and at least 10-fold higher than the reported SOD nanozymes under the same test conditions. We fully identified the active sites as well as addressed the catalytic mechanism of this activity. These findings should be of significance for the study on SOD-like nanozymes.

Table S1. Comparison of the typical parameters of SOD-like activities of C-dot nanozyme with the reported typical SOD nanozymes and natural SOD.

Nanozyme	[E] / $\mu\text{g}\cdot\text{mL}^{-1}$	Assay kit	Specific activity / $\text{U}\cdot\text{mg}^{-1}$	Inhibition rate / %	Ref.
Pt NPs-PVP	20	WST-1 by Sigma-Aldrich	/	58.30	6
MFC-MSNs	300	WST-1 by Sigma-Aldrich	/	~80	7
CeVO ₄	40	WST-1 by Sigma-Aldrich	/	~100	8
CaPB	960	WST-8 by Beyotime	/	~100	9
Rh-PEG NDs	80	WST-8 by Beyotime	/	~93	10
N-PCNSs	400	WST-1 by Dojindo	/	~75	11
PtPB	20	WST-1 by Dojindo	/	~45	12
CeO ₂	20	WST-1 by Dojindo	/	~60	13
PB	50	WST-1 by Dojindo	/	~90	14
Fe ₃ O ₄ NPs	2000	WST-1 by Dojindo	5.65	~11	15
Cu-SAzyme	12.3	WST-1 by Dojindo	448.22	~61	16
MnPS ₃	5.0	WST-1 by Dojindo	721.21	~70	17
pero-nanozysome	8.3	WST-1 by Dojindo	1257	~73	18
natural SOD	3.03	WST-1 by Dojindo	4743.8	~79	This
C-dot SOD Nanozyme	2.60	WST-1 by Dojindo	10767	~87	work

[E]: The nanozyme concentration.

3. L29 - deeply investigated, the word “deeply” reads unnecessary.

Reply: Thank you very much for your careful reading. We have deleted ‘deeply’ in the revised manuscript. In addition, we also have thoroughly checked the manuscript to eliminate similar issues.

4. L50 - catalysts in biomedical applications. Enlist some with key examples and supporting references.

Reply: Thanks for your constructive suggestion, and we have added several key examples and supporting references about biomedical applications of natural enzymes in the revised manuscript. The detailed discussion was revised as follows.

Essential to all life processes, enzymes are highly efficient biocatalysts that accelerate chemical reactions in physiological conditions. Their high catalytic efficiency and strong substrate specificity make them ideal catalysts in biomedical applications. For example, horseradish peroxidase is frequently utilized in enzyme-based sensing for biomarkers¹, viruses², and bacteria³. Catalase, which catalyzes the decomposition of hydrogen peroxide into water and oxygen, could relieve the hypoxia of tumor microenvironment to improve the antitumor efficiencies of radiotherapy⁴, sonodynamic therapy⁵, and photodynamic therapy⁶. Superoxide dismutase has been used as a therapeutic protein in skin inflammations⁷, inflammatory arthritis⁸, lung diseases and pulmonary fibrosis⁹, and diabetic nephropathy¹⁰, etc.

5. L55 – Avoid redundant expressions, e.g., broad variety. To somehow, broad and variety present the same meaning in this sentence.

Reply: Thanks for the suggestion. We have thoroughly checked the manuscript to avoid similar issues.

6. L61-62 – Referencing is not right. For instance, Up to now, nanozyme research mostly revolves around oxidoreductase activities, including oxidase (OXD), peroxidase (POD), catalase (CAT), and superoxide dismutase (SOD). This sentence is supported later with Ref. 6,7. None of them coat all the above-mentioned enzymes.

Reply: We apologize for this omission, and we have now corrected it in the revised

manuscript as follows.

Up to now, nanozyme research mostly revolves around oxidoreductase activities, including oxidase (OXD)⁻¹⁵, peroxidase (POD)⁻¹⁶, catalase (CAT)⁻¹⁷, and superoxide dismutase (SOD)⁻¹⁸like activities. Among them, the structure-activity relationship and rational design of POD nanozymes have been extensively studied due to their great potential in disease diagnostics and therapy¹⁹, and the activity of POD nanozyme, such as single atom nanozymes²⁰, is now comparable to that of natural peroxidase enzymes.

7. L67 – Avoid vagueness. a series of nanozymes. Enlist some with key examples and supporting references.

Reply: Thank you for your helpful suggestion, and we have revised this content as follows.

Apart from POD nanozymes, the reactive oxygen species (ROS) producers in vivo²¹, nanozymes with SOD⁻²², CAT⁻²³, or glutathione peroxidase (GPx)⁻²⁴like activities have been used to remove harmful ROS for cytoprotection, anti-inflammation, or antitumor theranostics²⁵⁻²⁷.

8. L105 - Scheme 1. What is the basis of this scheme? Any analytical proof that confirms the given mechanism?

Reply: We appreciate the reviewer for the insightful question. **Scheme 1** represents a concise description of the manuscript to give the reader a quick visual impression of the essence of the manuscript. In this work, a carbon nanodot (C-dot) SOD nanozyme with an ultra-high catalytic activity of more than 10000 U/mg was synthesized *via* the wet-oxidation method. We investigated the catalytic mechanism through surface structure modification and theoretical calculations, revealing that the SOD-like activity relies on the hydroxyl and carboxyl groups of the C-dots for binding superoxide anions through hydrogen bonds and the carbonyl groups conjugated with π -system for electron transfer.

To avoid misunderstanding and make it better readable to broad readers, the caption of **Scheme 1** was revised as follows.

Scheme 1. Schematic illustration of C-dots with ultra-high SOD-like activity for ameliorating ischemic stroke. a Synthesis of C-dots SOD nanozymes through oxidation of activated charcoal in the mixture of nitric acid and sulphuric acid; A proposed surface state-related catalytic mechanism of C-dot SOD nanozyme, in which the hydroxyl and carboxyl groups of the C-dots bind with superoxide anions through hydrogen bonds and the carbonyl groups conjugated with π -system serve as electron transfer. **b** The C-dot SOD nanozymes protect neuron during ischemic stroke *in vivo* by reducing intracellular reactive ROS level.

9. The work is of primary importance with poor sampling size. The sample variability and reliability are low. More than half of the work is based on a single sample characterization. For example, the data given in the supplementary files lacks graphite powder, carbon black, and activated charcoal. Reconsider all results data and make sure to present the analysis of all samples in a consistent manner.

Reply: Thanks very much for the constructive suggestion. In fact, all the analyses of samples in this study were at least performed 3 times or triple duplications. We have revised all the figures to show the data point in the figures for better understanding and assessment by readers.

In this work, graphite powder, carbon black, and activated charcoal were used as the raw materials to prepare C-dot SOD nanozymes. As suggested, we have added systematically analyses on these materials and other control samples (shown as revised Fig. 1e~f, Supplementary Fig. 3, and Supplementary Fig. 4) in revised manuscript as follows:

To investigate the determining factor for the SOD activity of C-dots, the surface structural differences of the C-dots derived from the three kinds of materials were investigated. The Raman spectra (Fig. 1e) of these C-dots displayed the G- (1596 cm^{-1}) and D- (1380 cm^{-1}) bands with I_D/I_G of 0.9 to 1.0, indicating large portion of defects on their surface induced by the strong oxidation. The XRD patterns of these C-dots are shown in Fig. 1f. The diffraction peaks of activated charcoal- and carbon black-derived C-dots with 2θ values of $25\sim 26^\circ$ and $42\sim 46^\circ$ attributing to the (002) and (100) facets, respectively, of graphite [powder diffraction file (PDF Card No. 01-0640)]. The (002) and (100) facets correspond to the facets parallel and perpendicular to the sp^2 -carbon layer of graphite, respectively, consisting with the TEM results. In contrast, no obvious diffraction peaks were detected in graphite powder-derived C-dots may due to the severe structural damages during the oxidation process. C 1s X-ray photoelectron spectroscopy (XPS) was conducted to semi-quantitatively analyze the surface structures of these C-dots (Figs. 1g-i). The XPS results indicated the presence of C=C, C-O, C=O, and O-C=O on the surface of these C-dots. The carbon-to-oxygen ratios of graphite powder-, and carbon black-derived C-dots were 1.33 and 1.45, respectively, much lower than that of activated charcoal-derived C-dots (2.01), indicating that graphite powder-, and carbon black-derived C-dots possessed higher degree of surface oxidation. The C=C content of activated charcoal-derived C-dots was as high as 71%, while those of graphite powder- and carbon black-derived C-dots were only 57% and 64%, respectively (Table S2). The high content of C=C implies a large π -electron system that could promote electron transfer and stabilize the intermediate products containing unpaired electrons. Therefore, sufficient C=C content is necessary for C-dots with high SOD enzymatic activity. In addition, we also found that compared to activated

charcoal-derived C-dots, carbon black-derived C-dots and graphite powder-derived C-dots exhibited lower carbonyl content, suggesting that carbonyl groups may also affect the SOD-like activity of C-dots. In the FT-IR spectra of these C-dots (Fig. 2d, and Supplementary Figs. 2 and 3), the strong bands at 3412, 1726, and 1240 cm^{-1} were ascribed to the stretching vibration of O-H, C=O, and C-O, respectively. The absorption bands at 1620 and 1350 cm^{-1} could be attributed to the stretching vibration of C=C and the bending vibration of C-H, respectively. The peaks ranging from 2870 to 2980 cm^{-1} were attributed to the stretching vibration of C-H in aliphatic hydrocarbons while the broadband around 2560 cm^{-1} was attributed to hydrogen bond stretching vibration.^{38, 56, 57} The significant difference in absorption peak intensities of these functional groups among the three kinds of C-dots indicated the difference in the contents of functional groups. Surface functional groups of C-dots could be quantified by $^1\text{H-NMR}$ spectroscopy using potassium biphthalate (PBP) as an internal standard⁵⁶ (Fig. 2e and Supplementary Fig.4). The total content of the reactable carboxyl and hydroxyl groups on graphite powder-, and carbon black-derived C-dots were calculated to be 0.85 and 3.08 mmol/g, respectively, lower than that of activated charcoal-derived C-dots (4.35 mmol/g). However, XPS results showed that the contents of C-O and O-C=O of graphite powder-, and carbon black-derived C-dots were higher than that of activated charcoal-derived C-dots, not consisting with the quantitative analysis of $^1\text{H-NMR}$. The less reactivity of C-O and O-C=O on the surface of graphite powder-, and carbon black-derived C-dots may be due to their large steric effect or other existence forms, such as ethers and esters, which could not be distinguished with hydroxyl and carboxyl, respectively, by XPS.

From the above results, it is reasoned that surface oxygen-containing groups play a key role in the catalytic activity of C-dots. The oxidation etching in the synthesis process destructs the relatively complete π -electron system of the original carbon materials, inducing oxygen-containing functional groups, such as carboxyl, hydroxyl, and carbonyl groups, on the surface of the C-dots. Simultaneously, the initially ordered sp^2 network structure of raw materials converted into the sp^2 - sp^3 hybrid nanostructure.

Finally, the small size (~2 nm) and large specific surface area enable C-dot SOD nanozyme to provide abundant binding and catalytic sites for the catalytic reaction. Oxygen-containing functional groups would combine with superoxide anions through weak interactions such as static electricity, hydrogen bonds, and other van der Waals forces, etc., facilitating redox reactions. Moreover, activated charcoal-derived C-dots synthesized with 0.5 h and 2 h reaction times exhibited lower SOD-like activities than the optimal 1.5 h (Supplementary Fig. 5), suggesting the reaction time in synthesis affected the SOD-like activity of the prepared C-dot nanozyme. Our previous work demonstrated that the surface-oxidation degree of C-dots increases as the reaction time prolong³⁴, which confirmed the surface-related SOD-like activity of C-dot nanozymes.

Fig. 1 Raman spectra (e) and XRD pattern (f) of C-dots prepared from graphite powder, carbon black, and activated charcoal.

Supplementary Fig. 3 FT-IR spectra of C-dots prepared from graphite powder, carbon black before and after reaction with PS.

Supplementary Fig. 4 ^1H NMR spectra of C-dots prepared from graphite powder, carbon black before and after reaction with PS.

10. What were the control samples? Any data for positive control/negative control?

Reply: For the SOD-like activity assay, the systematic positive control is the natural SOD enzyme (Siyomicro Bio-Tech). Regarding the mechanism study of C-dots, the original C-dot SOD nanozyme with the specific activity of 1.1×10^4 U/mg was the control, and we focused on the SOD-like activity changes after structural regulations.

Regarding the SOD-like activity assay, we tested the SOD-like activity of C-dots by using a Total Superoxide Dismutase Assay Kit (Dojindo Molecular Technologies), in which the highly water-soluble tetrazolium salt, WST-1, reacts with superoxide anions to form a water-soluble dye with optimal absorbance at 450 nm. The ratio of WST-1 reduced by superoxide anions is linearly related to the activity of xanthine oxidase and is inhibited by SOD. As shown in **Fig. R1**, the inhibition reaction by SOD in the red region occurs first, and the WST-1 reaction in the blue region can only occur after the SOD reaction is completed.

Fig. R1. The detection mechanism of SOD activity (manufacturer's instructions of S311-10, Dojindo Molecular Technologies).

Therefore, IC50 (50% inhibition concentration) of SOD or SOD analogs can be determined by colorimetry. In this test, there are three control groups (**Table R1**): (1) a blank control without inhibitors, (2) a sample blank control, and (3) a blank reagent control.

Table R1. Dosage of sample and control groups.

	Sample	Control 1	Control 2	Control 3
Sample solution	20 μ L		20 μ L	20 μ L
Ultrapure water		20 μ L		
WST working solution	200 μ L	200 μ L	200 μ L	200 μ L
Dilution buffer			20 μ L	20 μ L
Enzyme working solution	20 μ L	20 μ L		

The inhibitor rate of SOD (%) is calculated according to **Equation 1**:

$$\text{The inhibitor rate of SOD (\%)} = \frac{(A_{\text{Control 1}} - A_{\text{Control 3}}) - (A_{\text{Sample}} - A_{\text{Control 2}})}{(A_{\text{Control 1}} - A_{\text{Control 3}})} \times 100 \quad \text{Equation 1}$$

In this kit, 1U is defined as the amount of enzyme required in 20 μ L sample solution capable of inhibiting the reduction of 50% WST-1 and superoxide anions. The inhibitor rate (%) and enzyme activity (U/mg) we provided in this work were calculated according to the manufacturer's instructions. The data for the three controls were not showing in the article. Nevertheless, we are pleased to provide a table of raw data (**Table R2**) of the SOD-like activity of C-dots.

Table R2. The raw data of the SOD-like activity of C-dots.

Control 1: 0.852; 0.832; 0.827	Control 2: 0.043
Sample (0.08138 $\mu\text{g/mL}$): 0.298; 0.303; 0.302	Control 3 (0.08138 $\mu\text{g/mL}$): 0.288
Sample (0.16276 $\mu\text{g/mL}$): 0.207; 0.181; 0.18	Control 3 (0.16276 $\mu\text{g/mL}$): 0.165
Sample (0.325521 $\mu\text{g/mL}$): 0.127; 0.126; 0.127	Control 3 (0.325521 $\mu\text{g/mL}$): 0.103
Sample (0.651042 $\mu\text{g/mL}$): 0.117; 0.114; 0.116	Control 3 (0.651042 $\mu\text{g/mL}$): 0.074
Sample (1.302083 $\mu\text{g/mL}$): 0.158; 0.16; 0.169	Control 3 (1.302083 $\mu\text{g/mL}$): 0.059
Sample (2.604167 $\mu\text{g/mL}$): 0.227; 0.237; 0.246	Control 3 (2.604167 $\mu\text{g/mL}$): 0.05
Sample (5.208333 $\mu\text{g/mL}$): 0.347; 0.349; 0.353	Control 3 (5.208333 $\mu\text{g/mL}$): 0.047
Sample (10.41667 $\mu\text{g/mL}$): 0.476; 0.502; 0.516	Control 3 (10.41667 $\mu\text{g/mL}$): 0.045
Sample (20.83333 $\mu\text{g/mL}$): 0.617; 0.613; 0.605	Control 3 (20.83333 $\mu\text{g/mL}$): 0.044
Sample (41.66667 $\mu\text{g/mL}$): 0.699; 0.70; 0.704	Control 3 (41.66667 $\mu\text{g/mL}$): 0.043

11. Figure 1 (d-f) can be merged as a single image.

Reply: As suggested by the reviewer, **Fig. 1** has been revised as follows.

Revised Fig. 1. Characterization of C-dot SOD nanozyme. TEM (Inset: HR-TEM)

images of C-dots prepared from **a** graphite powder, **b** carbon black, and **c** activated charcoal. **d** The SOD-like activities of C-dots prepared from graphite powder, carbon black, and activated charcoal. **e** Raman spectra and **f** XRD patterns of C-dots prepared from graphite powder, carbon black, and activated charcoal as indicated. **C** 1s high-resolution XPS spectra with identification of peaks by curve fitting of C-dots prepared from **g** graphite powder, **h** carbon black, and **i** activated charcoal.

12. Table 1 is superficial, and many cells are empty. Authors cannot define superiority/inferiority with this little data. Move this into supplementary data and extend this through comparison.

Reply: We thank the reviewer for this comment, and we have now moved this table into supplementary data as **Table S1**. Meanwhile, we added more data to expand the table as follows.

Table S1. Comparison of the common parameters of SOD-like activities of C-dot nanozyme with the reported typical SOD nanozymes and natural SOD.

Nanozyme	[E] / $\mu\text{g}\cdot\text{mL}^{-1}$	Assay kit	Specific activity / $\text{U}\cdot\text{mg}^{-1}$	Inhibition rate / %	Ref.
Pt NPs-PVP	20	WST-1 by Sigma-Aldrich	/	58.30	6
MFC-MSNs	300	WST-1 by Sigma-Aldrich	/	~80	7
CeVO ₄	40	WST-1 by Sigma-Aldrich	/	~100	8
CaPB	960	WST-8 by Beyotime	/	~100	9
Rh-PEG NDs	80	WST-8 by Beyotime	/	~93	10
N-PCNSs	400	WST-1 by Dojindo	/	~75	11
PtPB	20	WST-1 by Dojindo	/	~45	12
CeO ₂	20	WST-1 by Dojindo	/	~60	13
PB	50	WST-1 by Dojindo	/	~90	14
Fe ₃ O ₄ NPs	2000	WST-1 by Dojindo	5.65	~11	15
Cu-SAzyme	12.3	WST-1 by Dojindo	448.22	~61	16
MnPS ₃	5.0	WST-1 by Dojindo	721.21	~70	17
pero-nanozysome	8.3	WST-1 by Dojindo	1257	~73	18
natural SOD	3.03	WST-1 by Dojindo	4743.8	~79	This
C-dot SOD Nanozyme	2.60	WST-1 by Dojindo	10767	~87	work

According to previous literature (*Analyst*, **146**, 1872 (2021)), there are 5 common methods used to test SOD-like activity of nanozymes: (1) HE probe, (2) NBT probe, (3)

WST probe, (4) INT probe, and (5) EPR measurement. In our manuscript, we have used three classical methods (above 2, 3 and 5) commonly used in most literatures to test the SOD-like activity of C-dots. Due to most literatures only provided data of one detection method or even only a relative trend chart, so some cells in the table are empty. Nevertheless, the good SOD-like activity of our C-dots can be demonstrated by comparing the inhibition rate of the specific activity.

13. Figure 2 - (d) FT-IR, what is a point in highlighting the overexpressed broad peak in the range 3200-3500 and 2750-2250 cm^{-1} ? What authors can explain the main peaks between 2000 and 1000 wavenumbers? Major peak numbers should be assigned with representative functional groups.

Reply: We thank the reviewer for this comment. We carefully re-analyzed the FT-IR spectra, and added descriptions of major peak numbers that should be assigned to representative functional groups. The detailed discussion has been added in SI as follows.

In this work, we performed a series of chemical modifications to transform the functional groups, especially hydroxyl and carboxyl groups, which involved changes in FT-IR. In order to accurately identify the absorption O-H, we carried out strict drying treatment on the samples, so that the O-H on the surface of the C-dots can be determined by the absorption band around 3400 cm^{-1} . The broadband around 2560 cm^{-1} could be attributed to the stretching vibration of the hydrogen bond of associating carboxyl group, which enabled one to distinguish a carboxylic acid from all the other carbonyl compounds⁵. For verifying the existence of carboxyl groups on the surface of C-dots, the sample was acidified or alkalinized by HCl or NH_4OH before FT-IR measurement. As shown in Supplementary Fig. 2, the peaks at 3400, 2560, 1720, and 1240 cm^{-1} in the FT-IR spectrum of C-dots-HCl increased significantly. After reacted with NH_4OH , the peaks at 2560, 1720, and 1240 cm^{-1} disappeared. Two strong peaks at 1600 and 1400 cm^{-1} appeared, which according to asymmetric and symmetric stretching vibrations of $-\text{COO}^-$, respectively, because the carboxyl group ($-\text{COOH}$) was converted to carboxylate anion ($-\text{COO}^-$). Therefore, we can attribute the peaks at

1720, 1240, and 2560 cm^{-1} to the C=O, C–O, and associated hydrogen bond, respectively, of carboxylic acid. While, the peak at 1620 cm^{-1} is more likely to be attributed to the acid-insensitive C=C.

Supplementary Fig. 2 FT-IR spectra of C-dots (untreated), C-dots-HCl (acidified), and C-dots-NH₄OH (alkalified).

14. Results – Data are not statistically shown; thus, any results could not be interpreted well. The figures and tables not showing the statistical differences. The author should perform the statistical analysis by Tukey or Duncan test and indicate the significance in the superscript (a, b, c or in asterisk, *, **, **) of each value.

Reply: We appreciate this suggestion. We have now added a statistical analysis to each value to be analyzed and indicated it with an asterisk. **Fig. 2** and **6** have been revised as follows.

Revised Fig. 2. Surface modifications to determine the catalytic active site of C-dot SOD nanozymes. a Illustration of C-dots modification. **b** SOD-like activity change of C-dots before and after passivation, reduction, and re-oxidation. (data represent means \pm s.d., $n=3$, * $P < 0.05$, ** $P < 0.001$, **** $P < 0.0001$, one-way ANOVA Tukey's multiple comparisons test). Source data are provided as a Source Data file. **c** FT-IR, **d** ¹H NMR, and **e** XPS spectra of C-dots, C-dots-PS and C-dots-PS-HCl. **f** C 1s high-resolution XPS spectra with identification of peaks by curve fitting of **f** C-dots-NaBH₄ and **g** C-dots-NaBH₄-HNO₃. **h** ¹H NMR spectra of C-dots-NaBH₄ and C-dots-NaBH₄-HNO₃.

Revised Fig. 6. Brain protection of C-dot SOD nanozymes against middle cerebral artery occlusion (MCAO)-induced ischemic and reperfusion injury. a Half-life analysis of C-dot SOD nanozymes in mice. **b** *Ex vivo* fluorescence imaging analyses of the accumulations of Cy5.5 labeled C-dot SOD nanozymes in the brains of sham (24 h post-injection) and MCAO mice (2 h, 6 h, and 24 h post-injection), and corresponding brain sections. **c** The cerebral infarcted area analyses of MCAO mice treated with different dosages of C-dot SOD nanozymes for 24 h (n = 3). **d** Representative 2,3,5-triphenyltetrazolium chloride-stained brain sections of MCAO mice treated with different C-dot nanozymes. **e** Quantification of cerebral infarct areas of MCAO mice treated with different C-dot nanozymes for 24 h (n = 3). **f** Neurological score analyses of the MCAO mice treated with different C-dot nanozymes for 24 h (n = 5). **g**

Representative images of TUNEL staining in the brain sections (scale bar = 25 μm), **h** malondialdehyde (MDA) assay in the brain homogenate, and ELISA assay of inflammatory factors **i** TNF- α , **j** IL-1 β and **k** IL-6 of the infarcted brain of MCAO mice treated by different C-dot nanozymes (n = 3). In c, e, f, h, i, j, k, data represent means \pm s.d. from 3 or 5 (as indicated) independent replicates (n.s., no significance, * $P < 0.05$, ** $P < 0.001$, *** $P < 0.001$, **** $P < 0.0001$, one-way ANOVA Tukey's multiple comparisons test). Source data are provided as a Source Data file.

General Comments:

Your manuscript has FIVE corresponding authors. Multiple corresponding authors are not encouraged. Please provide a detailed explanation about why this manuscript needs FIVE corresponding authors. Please kindly note only the authors who have solid contributions to this manuscript can be listed as corresponding authors.

Reply: We appreciate the reviewer's comment on this matter. While we agree that five corresponding authors is rare to one research article, in our case, we believe that it is justified for the following reasons. This work is an interdisciplinary study that covers material science, analytical chemistry, theoretical calculation, and nanomedicine with the collaboration between four labs spanning multiple years. This study couldn't have been completed without the collaborative work and expertise of all the corresponding authors. Prof. Fan and Prof. Liu conceptualized this work. The nanozyme theoretical calculations, cell and animal experiments in this work were performed in Prof. Yan and Prof. Fan's lab under their supervision. Collaborating with Prof. Pang, Prof. Liu established a series of surface modification methods to study the structure-property relationship of C-dots, and performed work on the investigation of C-dot SOD nanozymes. Currently, Prof. Liu is working in Prof. Zhang's lab. The work on the synthesis, surface modifications, and catalytic mechanism of C-dot SOD nanozymes were performed under the supervision of Prof. Pang, Prof. Zhang, and Prof. Liu. Therefore, Prof. Yan, Prof. Fan, Prof. Pang, Prof. Zhang, and Prof. Liu are listed as corresponding authors.

We have already stated the author's contribution as the journal requested.

REVIEWERS' COMMENTS

Reviewer #1 (Remarks to the Author):

The authors designed a carbon nanodot (C-dot) SOD nanozyme with ultra-high catalytic activity, and further investigations in terms of the C-dot SOD nanozymes including the potential to protect neuron cells during ischemic stroke were presented. I am much impressed by the SOD-like activity of C-dots that is several-fold higher than natural SODs. The authors have addressed the issues concerned with satisfaction, and the revisions are highly improved. So, I recommend Nat Comm to accept it as its present form.

Reviewer #2 (Remarks to the Author):

The revised version reads well. Authors have addressed all the comments raised in the last review. This manuscript can now be accepted for publication.

Responses to the reviewers' comments:

REVIEWERS' COMMENTS

Reviewer #1 (Remarks to the Author):

The authors designed a carbon nanodot (C-dot) SOD nanozyme with ultra-high catalytic activity, and further investigations in terms of the C-dot SOD nanozymes including the potential to protect neuron cells during ischemic stroke were presented. I am much impressed by the SOD-like activity of C-dots that is several-fold higher than natural SODs. The authors have addressed the issues concerned with satisfaction, and the revisions are highly improved. So, I recommend Nat Comm to accept it as its present form.

Response: We thank the reviewer for the positive comment and valuable suggestions, which helped us improve our manuscript.

Reviewer #2 (Remarks to the Author):

The revised version reads well. Authors have addressed all the comments raised in the last review. This manuscript can now be accepted for publication.

Response: We appreciate the reviewer's positive comment and valuable suggestions, which helped us improve our manuscript.